

# Mapping soil micronutrient concentration at national-scale: an illustration of a decision process framework

Christopher Chagumaira[1,2,3,4], Joseph G. Chimungu[4], Patson C. Nalivata[4], Martin R. Broadley[2,3], Madlene Nussbaum[5], Alice E. Milne[3], and R. Murray Lark[1,2]

[1]Future Food Beacon of Excellence, University of Nottingham, Sutton Bonington Campus, Loughborough, LE12 5RD, United Kingdom
[2]School of Biosciences, University of Nottingham, Sutton Bonington Campus, Loughborough, LE12 5RD, United Kingdom
[3]Rothamsted Research, Harpenden, AL5 2JQ, United Kingdom
[4]Lilongwe University of Agriculture and Natural Resources, Bunda College, P.O. Box 219, Lilongwe, Malawi
[5]Bern University of Applied Sciences (BFH), School of Agricultural, Forest and Food Sciences (HAFL), Switzerland

**Correspondence:** Christopher Chagumaira (christopher.chagumaira@nottingham.ac.uk; chris.chagumaira@gmail.com)

**Abstract.** Mineral micronutrient deficiencies (MND), prevalent in many countries, are linked to soil type. Stakeholders in Malawi, with different information needs, require spatial information about soil micronutrients in order to design efficient interventions. These stakeholders require reliable evidence for them to act, in most cases the outcome of their decisions involves financial costs and implications for farmers' livelihoods, food security and public health. They would not want to intervene
where it is unnecessary to do so or not fail to intervene where it is needed. Information about the concentration of micronutrient in soil is needed by stakeholders for decision-making. In practice this information is uncertain. Geostatistical methods and those based on algorithmically driven machine learning (ML) generate predictions of soil properties with measures of uncertainty, these measures are rarely linked to the decision-making process for which spatial information is required and it may not be clear to the stakeholders how to make use of the uncertainty information in decision-making. In this study we start from an
analysis of how stakeholders, in Malawi, may use uncertain spatial information to support decisions, providing the decisions about the acceptable quality of the information and how it should be collected. We then use this analysis as a framework to compare options for spatial prediction of micronutrients in soil by ML (e.g. random forest) and geostatistical methods (e.g. linear mixed models).

## 1 Introduction

Mineral micronutrient deficiencies (MND) prevalent in many countries, including Malawi, are linked to soil type (Hurst et al., 2013; Joy et al., 2015; Gashu et al., 2021), therefore the concentration of micronutrients in staple cereals is spatially dependent (Gashu et al., 2020). This is of particular concern in countries where diets are sourced locally as there is a greater risk of deficiency. For example, Gashu et al. (2020), showed that the concentration of Se in grain teff (*Eragrostis tef* (Zucc.) Trotter) was dependent on soil properties such as pH and organic carbon content. Joy et al. (2015) showed that there is strong evidence that
cereals grown in Malawi have restricted uptake of micronutrients and dietary mineral intake is influenced by mineral content



of edible portions of crops. Studies done in Ethiopia and Malawi have shown positive relationships between the concentration of Se in grain and Se-biomarker values in women of reproductive age (Phiri et al., 2019; Belay et al., 2020).

Spatial information about soil properties is needed to design site-specific interventions to address MND such as the promotion of practices like agronomic biofortification with additions of micronutrients to fertilisers (Botoman et al., 2022; Joy et al.,
2022). Soil properties cannot be measured everywhere, and to map the variation in a soil property one must interpolate from measurements made on samples taken at a number of locations across the area of interest (Webster, 1977). However, the resulting predictions are uncertain due to the inherent variation of soil at multiple scales and resulting sampling error, measurement error and uncertainty arising from predictive factors in our spatial models.

Spatial information about the soil can be derived from soil survey. Conventionally, a soil survey was almost always based
on classification of the soil, and the delineation of map units which are each identified with one soil class, or an association of soil classes, as explained in the map legend (Dent and Young, 1981). Information on the soil was organized with respect to the classes, e.g. estimation of class means for soil properties, or provision of a description and analyses of a 'representative profile'.

The soil surveyor would usually draw sharp boundaries between the map units they recognise (Webster, 2015). The variation
of soil properties within each mapping unit is treated as an independent and identically distributed random variable for purpose of quantifying uncertainty (Webster and Beckett, 1968; Webster and Lark, 2013). However, the implicit model of spatial variation–sharp boundaries between map units– is not adequate to fully capture soil variation. One might expect more efficient predictions or similar map accuracy with fewer samples (Nussbaum et al., 2018) from a model in which soil variation occurs continuously and at multiple spatial scale in space. Such a model is provided by the regionalised variable theory of Matheron
(1965) which underpins geostatistical methods. The pioneering work of Burgess and Webster (1980) introduced this methodology to soil science. The approach has been taken up and developed substantially over the intervening period (Malone et al., 2018) leading to the development of digital soil mapping (DSM, McBratney et al., 2003).

Geostatistical methods aim to capture the spatial dependence, by treating soil variation as an outcome of a random process (Webster, 2000), through predicting soil classes or properties onto grid points or cells (raster in Geographic Information System
terms) from a set of point observations which might be on a systematic grid, or assembled from past surveys with different designs. Additional points near the grid nodes (close pairs) are needed to give reliable estimates of the fine-scale covariance in the soil property (captured formally in the variogram model) for geostatistical methods (Webster and Lark, 2013; Lark and Marchant, 2018). The simplest geostatistical method, ordinary kriging (OK), uses only data on the target soil property and entails the assumption that its unknown mean value is locally constant. However, the assumption can be relaxed by modelling
the mean as a function of covariates. These might just be coordinates, to capture a simple trend (universal kriging) or could include other variables such as remote sensor data (kriging with an external drift). All the kriging approaches can be regarded as forms of the Empirical Best Linear Unbiased Predictor (E-BLUP, Webster and Oliver, 2007). The E-BLUP is based on the linear mixed model (LMM) with covariates as the fixed effects, spatially correlated random effects and uncorrelated residuals.

More recently, DSM practitioners have turned their attention to machine learning (ML) methods for spatial prediction of soil
properties. Machine learning algorithms refers to a large class of data-driven algorithms originally developed for data mining





and pattern recognition. Most ML methods do not assume a pre-defined functional form of the response-covariate relationship and therefore do not require or have relaxed assumptions of the model errors to follow a pre-specified distribution (Arrouays et al., 2020). Digital soil mapping by ML methods can use the same covariates as LMM for spatial prediction with the E-BLUP. For spatial prediction with OK, sample points are best distributed on a grid or some other design which achieves spatial

coverage (de Gruijter et al., 2006). If covariates are incorporated through an E-BLUP then the estimation of fixed effects coefficients must also be considered in the design of sampling. Brus et al. (2006) showed how requirements for estimating both the fixed and random effects components of the E-BLUP influence an optimal sample design. If spatial prediction does not directly exploit spatial dependence, the sample selection may be based on the variation of the covariates. For example, conditioned Latin hypercube sampling (Minasny and McBratney, 2006) aims to spread the sample points over the covariate

space. Ma et al. (2020) suggested that the feature space coverage sampling design is optimal for ML because it covers the multivariate covariate space equally. Once soil data are collected the prediction model must be built. The incorporation of covariates which are poor predictors may inflate the uncertainty of the final prediction. In geostatistics, a variable selection procedure may be used while ML methods aim to weight all covariates appropriately.

Variability, sampling effort and modelling of the relationship between soil and covariate all contribute to the uncertainty

in spatial predictions. Most studies in pedometrics provide some measure of uncertainty alongside spatial predictions, but all too often these are done in a "vacuum" without considering the particular requirements of a specific end-user in mind. The question therefore usually remains open whether an advance has been achieved from the perspective of the user of information (Lark et al., 2022). Although attempts have been made to quantify and communicate uncertainty (Chagumaira et al., 2021), pedometricians have realised that these measures are rarely linked explicitly to the decision-making process for which spatial

information is required (Wadoux et al., 2021) and it may not be clear to the stakeholders how to make use of the uncertainty information in decision-making. With increasing technological advances, there is a demand to provide soil information from new soil surveys and using legacy data. We aim to analyse how a 'decision process' can be used to address various information needs by different stakeholders. This has implications for how good predictions have to be, i.e. the trade-off between sampling effort and degree of precision. This study fits well in the research agenda focusing on the decision process of key users of soil

information to quantify impacts of uncertainty in spatial information raised by Wadoux et al. (2021) among the ten challenges for future of pedometrics.

## 2   Decision process

Spatial information about environmental variables is required to serve stakeholders with different needs. These stakeholders require reliable evidence for them to act, in most cases the outcome of their decisions may involve financial costs and impli-

cations for farmers' livelihoods, food security and public health. They would not want to intervene where it is unnecessary to do so or not to intervene where it is needed. The needs of stakeholder are unsystematically reported in most pedometrics research, perhaps a lot of consideration of their needs go unaddressed. Without a full analysis of the implications of decisions end-users make, we cannot fully address their requirements. In this section we will discuss different clusters of decisions and

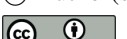


concepts, the 'decision process', that should be considered when planning for sampling and spatial prediction. The general

decision theory concepts which we use can be found in standard texts such as Peterson (2017).

## 2.1 Decisions and losses

As stakeholders have different information needs, it is important to consider the use of soil information. This can be characterised in terms of a set, I, of four questions about the information.

$$
I \begin{cases}
\text{(I1) What decision is to be made with the information?} \\
\text{(I2) At what management unit is the decision made (e.g. field, farm, district)?} \\
\text{(I3) How is soil information used in the decision?} \\
\text{(I4) What are the possible outcomes from the decisions given uncertainties?} \\
\text{(I5) What is the potential legacy value of the survey?}
\end{cases}
$$

There is a need to consider how stakeholders use the information to make decisions. Figure 1 is an illustration of how a stakeholder may use spatial information based on soil pH and texture to decide on liming rate. For each *decision* made, the *outcome* depends on the possible *states*. The *state* is the state of affairs which our soil information predicts. In this example the stakeholder has a choice of three actions to intervene (apply lime at one of two rates) or not (no lime application). In this example, the stakeholder needs to make a decision about liming at a specific management unit (e.g., farmer's field) and the

decision will be made using data on soil pH and texture. If a stakeholder decides to apply a greater amount of lime when there was no need, the losses attached would be the unnecessary costs (e.g. purchasing of lime and labour) and increased risk of immobilising micronutrients in the soil. When the stakeholder decides not to apply lime when there is a need for it, the loss attached to the decision would be yield loss. The stakeholder may try to be cautious due to uncertainties and decide to apply a moderate amount of lime when there is need for a greater application, the loss attached to this decision would be some yield

loss.

In our example of soil pH, the decision whether and how to intervene will reflect the stakeholder's assessment of the loss attached to each possible *outcome*. The loss is relative to an *outcome*. For example, one might attach a cost in monetary terms to yield loss from failing to lime when necessary, and opportunity cost of unnecessary liming. One may represent the costs as a continuous loss function (Ramsey et al., 2002; Lark and Knights, 2015). For a given state (e.g. optimal liming rate)

the loss is the cost of overestimation or underestimation of the soil variable(s) which determine(s) the state. If the loss from overestimation and underestimation by the same amount are equal then the loss function is symmetrical. This is unlikely in general. In the example here, if we underestimate fertiliser required by a certain quantity, we may expect a larger loss, due to yield reduction, than if we overestimate by the same amount. As a result the loss function is asymmetrical. From the perspective of environmental management the asymmetry may go either way– with more severe losses associated with over-application

that results in emissions from the field.





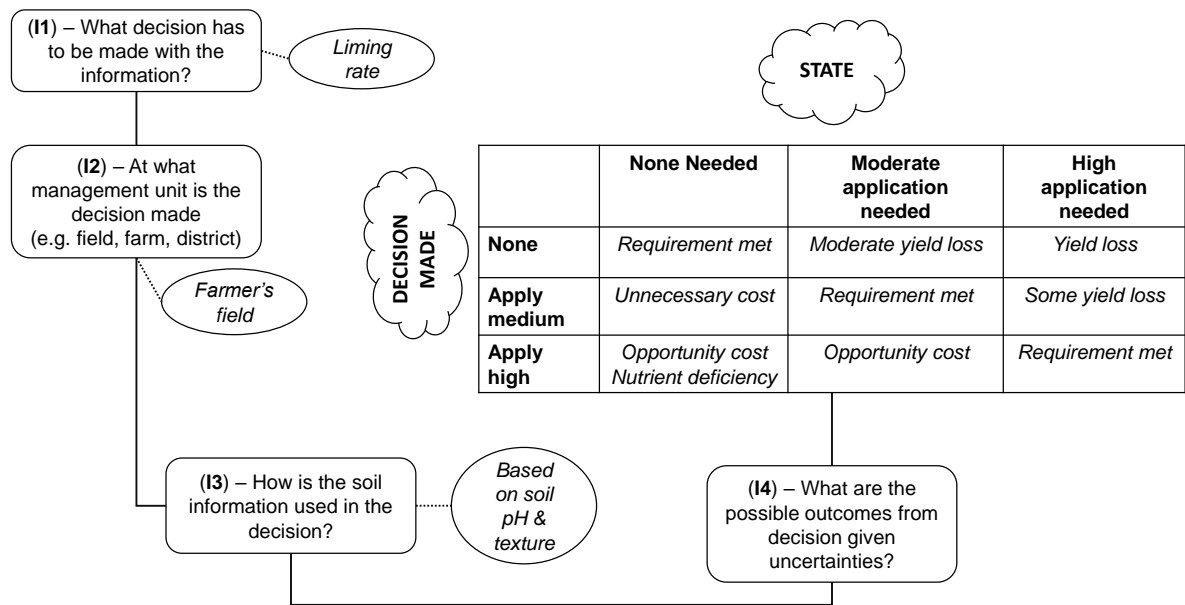

**Figure 1.** Illustration of a decision process for deciding on liming rate.

## 2.2 Stakeholders

The considerations described above highlight the importance of considering who the stakeholders are: who carries the risks from the outcomes, and who bears the costs of collecting the information. Three types of stakeholders can be defined: the information user (S1), the sponsor (S2) and the indirect or social client (S3). These three may have different soil information needs. The information user (S1) may include land managers, nutritionists, agronomists, soil scientists, policy-makers, environmental managers, governments and donors e.g. non-governmental organisations. Examples of sponsors (S2) are governments, research organisations or consortia e.g.G-BASE (Johnson and Breward, 2004), GlobalSoilMap.net (ISRIC - World Soil, 2009), GeoNutrition projects (GeoNutrution, 2017) and donors e.g. non-governmental organisations. The third group include farmers and the general public who suffer from the consequences of errors made by the group S1. At times, groups S1, S2 and S3 may have a common interest to a problem and in such a case we propose a "composite stakeholder", a user of information informed about the needs of the social client, and the constraints on the sponsor, and balancing them in a way that is socially and politically acceptable.





"Composite" Stakeholders (CS)
$\begin{cases} \text{(S1) Who makes the decision? (user)} \\ \text{(S2) Who pays for the survey? (sponsor)} \\ \text{(S3) Who is affected by the outcome? (social client)} \end{cases}$

The uncertainty in the predictions of the soil variable of interest can usually be reduced by increasing the number of samples taken. Many survey sponsors and users, who make decisions about information, (I1 to I5), which have an impact on sample size, have little knowledge of statistics and so might not be equipped to relate sample size to measures of uncertainty and, in turn, to implications for decision. In most cases their decision are based on financial costs of the surveys. Therefore, the surveyor needs to engage with stakeholders to identify sampling designs that maximise benefits of sampling over the cost by considering the questions (O1 to O6).

O
$\begin{cases} \text{(O1) What type of survey is appropriate?} \\ \text{(O2) What measure of uncertainty is required?} \\ \text{(O3) How should survey outcomes (including uncertainty) be communicated?} \\ \text{(O4) How many samples should be taken?} \\ \text{(O5) How should the samples be distributed?} \\ \text{(O6) Which covariates should be used in modelling?} \end{cases}$

In order to address some of the concerns raised by the questions above we need to consider the questions:

V
$\begin{cases} \text{(V1) Can we characterise the spatial variability of the soil properties of interest?} \\ \text{(V2) Can we characterise other sources of uncertainty? (e.g. analytical lab analysis)} \end{cases}$

These questions are concerned with how we capture spatial variation (V1) and how uncertainty is quantified (V2). Data and surveys are costly, and therefore rational decisions should be made in this respect. The resources questions (R1to R4) are concerned with budgets and information. R1 is important especially for the survey sponsor (S2) because this is where the decisions on sampling are made.

R
$\begin{cases} \text{(R1) Is budget fixed or negotiable? i.e. is there an actual decision about sampling?} \\ \text{(R2) Are there legacy sources of information?} \\ \text{(R3) Are there legacy data?} \\ \text{(R4) What covariates are available?} \end{cases}$

The key questions raised by information needs (I), stakeholder (S), spatial variation and uncertainty (V) and resources (R) are the "base level" questions about a situation where soil information is needed. The uncertainties associated with predictions



from partial sampling need to be quantified and communicated effectively to the stakeholders (S1 to S3) who need to make decisions based on their information needs (I1 to I5). First, the value of uncertain information to CS must be quantified as function of uncertainty (U1). Secondly, can the acceptable uncertainty be quantified (U2)? Finally, can the survey effort be linked to tolerable uncertainty measure? (U3).

$$
U \begin{cases} \text{(U1) Can the value of uncertain information to CS be quantified?} \\ \text{(U2) Can acceptable uncertainty be specified quantitatively?} \\ \text{(U3) Can survey effort be linked to the uncertainty measure?} \end{cases}
$$

U1 is a complex question which has received a good deal of attention, and this has been summarised by Lark et al. (2022). Lark et al. (2022), Lark and Knights (2015), Ramsey et al. (2002) and Giasson et al. (2000) discuss how value of information theory (Howard, 1966) can be applied to soil information. Lark and Knights (2015) give a simple example in which a loss function for a liming decision based on field-scale estimate by simple random sampling is used to calculate the expected loss of a decision based on an estimate as a function of its standard error (U1). This in turn can be expressed as a function of sample

size (U3) given the variance of pH within the field (V1), and so the marginal benefit (reduced expected loss) of an additional sample point may be calculated. In this setting a rational value for tolerable uncertainty (V2) could be the standard error at which the marginal reduction of the expected loss equals the marginal cost of an additional sample (Lark and Knights, 2015; Lark et al., 2022). Criteria for specifying acceptable uncertainty measure (V2) may be conventions based on experience such as purity values for soil maps specified in survey contracts (Western, 1978) or the offset correlation (Lark and Lapworth, 2013)

in which survey effort can be limited to a measure of the robustness of the final map to arbitrary variation of the origin of a survey grid (U3) on the basis of the variogram of the target variable (V1).

## 3   Case study

Malawi is a setting for much activity in the GeoNutrition project, which addresses the recalcitrant challenges of MND in SSA. The GeoNutrition project aims to examine whether better interventions to address MND could be based on spatial information,

rather than assuming that the same intervention is required everywhere (Gashu et al., 2020).

The 'Addressing hidden hunger trials (AHHA)' trial was conducted in Malawi to test the efficacy, for the alleviation of Se deficiency, of consuming maize flour enriched with Se by agronomic biofortification (Joy et al., 2019). The Se-enriched maize flour, and non-fortified flour, was provided to households in a randomized double-blind design. Comparison of biomarker measurements before and after a period, of 8 weeks, in which this flour was consumed were recorded for one woman of

reproductive age and one school-aged child in each household. It was found that, for households receiving the fortified flour, Se status of the individuals improved over the period, but did not change in the control group and that agronomic biofortification is a viable strategy to address Se deficiency (Joy et al., 2022).




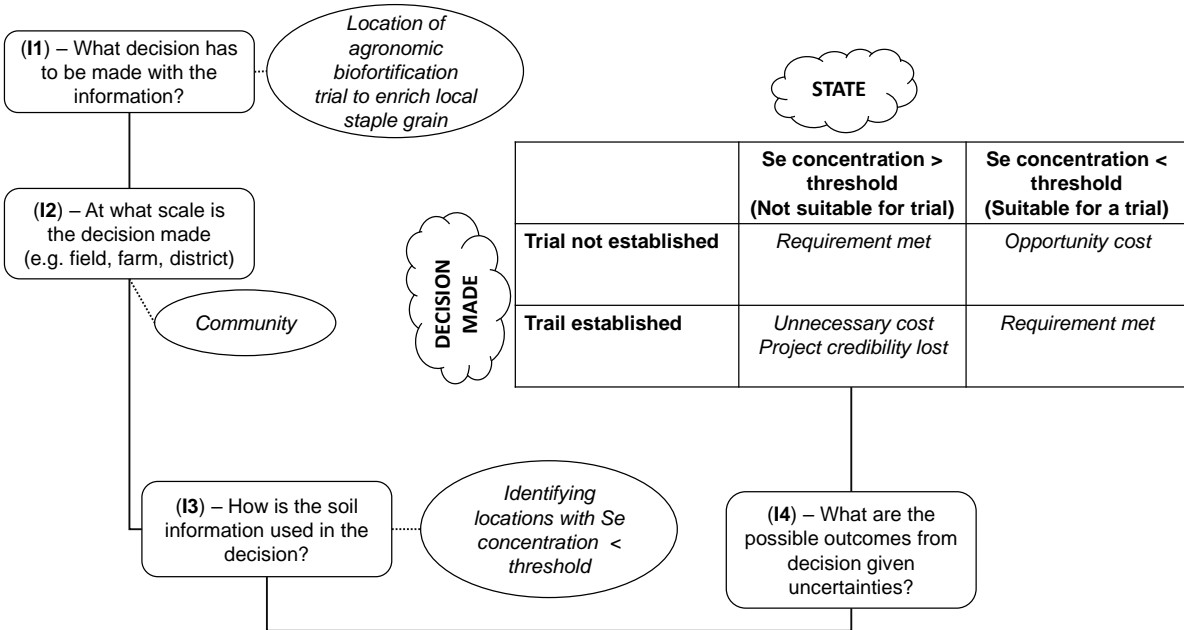

**Figure 2.** Summary for the decision process for deciding on location of trials, for agronomic biofortification, where concentration of $Se_{sol}$ is less than a threshold.

In this case study we consider how soil information could be used for the design of a further round of experiments to evaluate the potential of agronomic biofortification as a strategy 'at scale' with local agronomic fortification. In the AHHA trial the fortified maize was grown at a single central location. A key question is whether agronomic biofortification can be practiced by farmers, and whether this benefits local communities who then consume the produce. The team managing the experiment, agronomists and soil scientists, public health and nutrition specialists, want to use spatial information to identify potential sites where the concentration of soil soluble Se ($Se_{sol}$) is small. When a list of experimental sites has been produced, some initial engagement with each local community will be undertaken to explain the project. Informal consent will be obtained to sample soil from local fields to check that the concentrations are small. If they are, then a second more intensive phase of sensitisation will be undertaken leading up to the agronomic component of the trial and the feeding trial. If the proposed site does not have small Se concentrations in the soil then it will be abandoned and an alternative will be examined instead. This means that a 'false positive', a site incorrectly identified as having small Se supply from the soil, will not entail losses due to the completion of a trial where the effects of agronomic biofortification are small. However, it will entail costs due to the effort required for the initial community engagement, the completion of soil sampling, and the loss of goodwill and credibility with the community if the trial is not completed there. This might have implications for wider public attitudes to the trial, and further up-scaling of the approach in the future. A decision process for deciding on location of trials, for agronomic biofortification, where concentration of $Se_{sol}$ is less than a threshold are summarised in Figure 2.



Given this background, we can now consider the question sets in Section 2.1.

I1. The decision is where to locate $n$ community-level trials, given that the objective is to do them on sites which, among other considerations, have small $Se_{sol}$ concentrations.

I2. The decision is to be made at community level. Conventionally-produced and agronomicaly biofortified crops will be grown separately by recruited farmers in each community, the soils of which have small $Se_{sol}$ concentrations.

I3. Candidate communities will be those at locations where the predicted $Se_{sol}$ concentration is less than a threshold. At present there is no accepted threshold to define Se-deficient soil, but agronomists and soil scientists agreed to use the $25^{th}$ percentile of the measured $Se_{sol}$ concentration, from GeoNutrition samples (Gashu et al., 2021), as a threshold, denote by $Se_{threshold}$. Other factors will influence the decision (e.g accessibility), so the project team will weigh up the risk that the site is not suitable because of soil conditions against other factors when short-listing communities for participation.

I4. A community is added to the list for the trial because the predicted $Se_{sol}$ concentration is below a threshold. The soil will be sampled locally, so if the prediction is found to be correct (at least with respect to the threshold), then the site will be correctly included in the trial, and the process of community engagement discussed above will proceed smoothly. If the soil at the community is found to have Se concentration in excess of the threshold, then the site is not suitable for the trial. The effort already put into community engagement will be largely wasted, the withdrawal of the team may affect their credibility, and that of spatial soil information, in the eyes of the local community. They may be less willing to engage in similar trials in future, and resistant to future attempts to engage them in work to scale up agronomic biofortification practices (or maybe other campaigns to address MND). Uncertainties in the soil information may also mean that eligible sites might not be considered. This could result in opportunity costs for well-positioned locations, but given the size of the threshold this is unlikely to limit the completion of the trial.

I5. The proposed activity is not a survey to undertake mapping or to provide information, but is secondary data analysis. This question is therefore not relevant.

S1. The project team will make the decision, along with local officials.

S2. The sponsor is the Government of Malawi.

S3. In so far as communities are included, at least initially, in the trial but are found to be unsuitable, the project team wastes some resource. They also lose credibility, and this has some impact on the sponsor as well, who are associated with the trial. Local communities who lose faith in the promoters of the study may later choose to exclude themselves from activities, including up-scaling of MND interventions, from which they would have benefited.

O1. This is a secondary analysis, so no new survey decisions have to be made. Spatial predictions of $Se_{sol}$ concentration are needed effectively at point scale since communities are small relative to the sampled domain (all the country).





O2. A quantitative measure of uncertainty is needed, so that the project team can assess the risk that the site is unsuitable when making this judgement. As there is a threshold specified, the probability that the soil Se at the site is below the threshold would be a useful measure.

O3. Chagumaira et al. (2021) found that a wide range of stakeholders found the probability that the true value of a variable is below or above a threshold is an effective way to communicate uncertainty in spatial information. They found little evidence that different methods to express this probability worked better than others, but methods such as calibrated phrases (Mastrandrea et al., 2010; Lark et al., 2014) or maps of probability with pictographs at different candidate communities might be appropriate.

O4. This is secondary data analysis and this decision is not relevant.

O5. As this is secondary data analysis this decision is not relevant.

O6. The available covariates are surface slope, and topographic index mapping derived from the MERIT Digital Elevation Model of Yamazaki et al. (2017). Downscaled climate data was obtained from CHELSA data set (Karger et al., 2017). Average and variance annual net primary productivity, enhance vegetation index, normalised difference vegetation index and soil adjusted vegetation index were obtained from the MODIS remote sensor satellite (Justice et al., 1998) (see Table S1).

Questions under V and R: probably not relevant to a secondary data analysis.

U1. The costs of a false positive are partly tangible (time and resources wasted) and intangible (loss of goodwill and credibility). Costs of false negatives are harder to evaluate. Given the size of the threshold the project is unlikely to be short of communities. However, if the criterion is too strict we might miss out on communities good for other reasons (accessibility etc). Overall, cost of false positives exceeds that of false negatives.

U2. On the above basis a critical probability might be elicited from a group, following Chagumaira et al. (2022) such that a community is considered for inclusion in the trial of $\mathrm{P}\left(\mathrm{Se}_{\mathrm{sol}} < \mathrm{Se}_{\mathrm{threshold}}\right)$ exceeds that critical value. Considerations in U1 suggest that the loss function is asymmetrical with a larger loss from false positives than false negatives. On this basis we would expect that, if the probability is presented as $\mathrm{P}\left(\mathrm{Se}_{\mathrm{sol}} < \mathrm{Se}_{\mathrm{threshold}}\right)$, then the critical probability will be in excess of 0.5.

U3. Not relevant for a secondary data analysis.

## 4 Materials and methods

### 4.1 Data and study area

Details of soil sampling and laboratory analysis are given by Gashu et al. (2021). Soil soluble Se was extracted in 0.01M KNO$_3$. The objective of the field sampling in Malawi was to support spatial prediction of soil and crop micronutrient concentration.



The location of sample points were obtained by using the $k$-means methods as encoded in the spcosa package (Walvoort et al., 2010) for the R platform (R Core Team, 2021). This method allows one to form a sample which gives good coverage while incorporating the fixed prior points in the sample. There were 820 prior points from the 2015–16 micronutrient survey of Malawi (Phiri et al., 2019), and a further 890 spatial coverage points were added by using spcosa plus a further 190 'close-pair' sample locations. The close-pair samples are required to support spatial modelling–10% of the total samples (Lark and

Marchant, 2018). The additional points near the grid nodes (close pairs) are needed to give reliable estimates of variogram parameters and they increase spatial coverage to minimise kriging variances especially at the border of a study region. A total of 1,812 sites of grain and soil samples were taken. However, some sample location had positional uncertainties that were attributed to either poor satellite signal or enumerators not giving the devices enough time to establish the location and were not usable for further analysis. These six samples were removed from the spatial prediction of $Se_{sol}$. Of these 1806 sites, 10%

of the data was used to create an independent dataset for validation. The 190 close-pair sites were included in the dataset used for prediction and training of the models. From the data with the close-pairs removed, 160 points were selected into the validation dataset using simple random sampling without replacement.

### 4.2   Linear mixed models for spatial prediction

The best unbiased linear predictors (BLUP) is computed from a linear mixed model (LMM) is an additive combination of one

or more fixed-effects and one or more random effects. The independently and identically distributed component is in the model but does not affect the prediction. When the fixed effects are just an unknown constant mean then the BLUP is equivalent to ordinary kriging. The theory of LMM as a geostatistical model for spatial prediction is described in greater detail by Lark et al. (2006). The variance parameters are estimated by maximum likelihood or residual maximum likelihood (REML, Stein, 1999). Both maximum likelihood and REML are based on the assumption that random effects have a joint Gaussian distribution,

therefore it is important to study the descriptive statistics of the dataset and do the transformations when necessary. Estimation of variance parameters by REML eliminated bias that results from estimating semivariance computed from a variogram of residuals by using method-of-moments when there are fixed effects added to a constant mean.

Exploratory data analysis, using simple summary statistics and plots (e.g. Q-Q), were done to check whether transformation was needed to make the assumption of normality reasonable. Table 1 shows the summary statistics of $Se_{sol}$. Exploratory

analysis of the data indicated a possible spatial trend, but this was not pronounced, as indicated on the exploratory variograms. Two LMM were therefore considered as options. First, with a constant mean as the fixed effect, second with fixed effects selected from available coordinates (Table S1), including spatial coordinates.

### 4.2.1   Constant mean as the fixed effect

The $Se_{sol}$ data was strongly skewed, transformation to logarithm reduces this, but the histogram of the data stilled showed

marked non-normality and bimodal (see Table 1 and Figure S3). We therefore considered transformation of the data by Gaussian anamorphosis (GA) using Hermite polynomials computed with the anam.fit function for the RGeostats package (MINES ParisTech / ARMINES, 2022) for the R platform. Variograms were then estimated for the transformed data using the estimates



**Table 1.** Summary statistics of for soil soluble Se concentration (Se$_{sol}$), cross-validation errors for ordinary kriging and robust REML E-BLUP, and out-of-bag cross-validation errors for random forest.

|  | Concentration of Se$_{sol}$ ( µg kg$^{-1}$) | Ordinary kriging cross-validation errors | robust REML E-BLUP cross-validation errors | Random forest out-of-bag errors |
|---|---|---|---|---|
| Mean | 3.94 | 0.00 | -0.07 | -0.00 |
| Median | 3.30 | 0.01 | -0.02 | -0.05 |
| Standard Deviation | 2.99 | 0.55 | 0.76 | 0.50 |
| Minimum | 0.18 | -2.78 | -2.86 | -1.90 |
| Maximum | 18.8 | 3.11 | 2.48 | 2.41 |
| Skewness | 1.29 | -0.26 | 0.00 | 0.59 |
| Octile skewness | 0.24 | -0.03 | -0.16 | 0.14 |

due to Matheron (1962), Dowd (1984) and Cressie and Hawkins (1980). Exponential variogram model were fitted by weighted least squares and the models were validated by cross-validation. Following Lark (2000), we chose the variogram fitted by the different estimators that had a standardised squared prediction error (SSPE) falling within the 95% confidence interval around the expected value of 0.455. Predictions of the GA-transformed scale was obtained by ordinary kriging (OK) at validation sites and at points in a grid across Malawi. The value of Se$_{threshold}$, 1.49 µg kg$^{-1}$, was transformed to the GA scale (-0.681), and the probability that the true value is smaller than this was computed assuming a prediction distribution of the mean and variance equal to the OK estimate and kriging variance, respectively. Median-unbiased estimates of Se$_{sol}$ in units of µg kg$^{-1}$ were obtained by back transformation.

### 4.2.2 Fixed effect selected from available covariates

Exploratory analysis with spatial coordinates and environmental covariates (see Table 2) on fixed effects suggests that an assumption of normal random effects was plausible on transformed to natural logarithm, although with some outliers present. For this reason we estimated variance parameters for the LMM by using robust REML following Künsch et al. (2013).

Robust REML algorithm automatically identify outliers within a dataset and the outliers receive small weight when estimating model parameters (Nussbaum et al., 2012). The covariance matrix of the regression coefficients and the variogram parameters are estimated by georob (Papritz and Schwierz, 2021), for the R platform, either by REML or maximum likelihood from the Se$_{sol}$ dataset and values of the environmental covariates. The estimating equations are robustified by replacing the standardised errors by a bounded function of them and introducing a suitable bias correction terms for Fishers consistency in the Gaussian model (Künsch et al., 2013; Papritz, 2021). The bounded function of the residuals becomes the tuning parameter, $c$, of robust REML and is used to control the robustness of the procedure. The lower the value of $c$, the more the outliers are penalised by lower weights. The resulting predictions of Se$_{sol}$ were on a log-scale, and they needed to be back-transformed



to aide interpretation. We used the standard unbiased back-transformations for log-normal kriging procedure (Cressie, 2006) using the lgnpp function of georob package (Papritz and Schwierz, 2021).

Not all available covariates are useful for spatial prediction when using LMM. If all covariates are used for prediction some problems can be encountered. These include risk of propagating error in the regression coefficients when weak covariates are included in a model. Other problems may include overfitting, over-adaptation to the training data hence leading to a lack of generalization and poor predictive performance in new conditions. In order to address some of these challenges Lark (2017) suggested that the problem of variable selection is considered in terms of multiple hypothesis testing. Hence in this study,

covariate selection was done by using the method described in Lark (2017) in which false discovery rate is controlled with the $\alpha$-investment.

In this method of variable selection, a prior ranking of potential predictors of $Se_{sol}$ was required. The rankings were provided by a panel of eight plant and soil scientists from the University of Nottingham, Rothamsted Research, Lilongwe University of Agriculture and Natural Resources and Addis Ababa University. The rankings were based on a priori expectations of the order of importance based on the processes involved. The rankings from the panel was obtained through group elicitation facilitated

by a statistician. The selected order for testing potential predictors for $Se_{sol}$ obtained in the group are shown in Table 2.

**Table 2.** Sequence of predictors for $Se_{sol}$ concentration (environmental covariates) for testing with the $\alpha$-investment

| Order | Environmental Covariate |
|-------|-------------------------|
| 1 | Downscaled mean annual precipitation (BIO12) |
| 2 | Downscaled mean annual temperature (BIO1) |
| 3 | Slope (SLOPE) |
| 4 | Topographic index (TIM) |
| 5 | Average enhanced vegetation index (EVI) |
| 6 | MODIS band 7 (MB7) |
| 7 | MODIS band 2 (MB2) |
| 8 | MODIS band 1 (MB1) |
| 9 | MODIS band 3 (MB3) |

The climate variables (mean precipitation and rainfall) were as the most likely useful predictors because it is expected that rainfall and temperature can enhance the mineralization of organic matter in the soil thereby releasing Se bound in organic compounds into soil solution. The terrain variables, slope and topographic index where considered next. The MODIS enhanced

vegetation index (EVI) and bands were also considered because they measure vegetation vigour and health.

In a LMM framework, evidence that the coefficient of a covariate is significantly from zero can be tested by a Wald test

$$W_T = \frac{\left(\hat{\theta}_1 - \theta_0\right)^2}{\text{Var}(\hat{\theta}_1)}. \tag{1}$$



Where $\theta_1$ and $\theta_0$ are the vectors of all parameters estimated by REML, from fitting the model with additional fixed effect
and the simple model, respectively. The statistic is asymptotically distributed $\chi^2$ with one degrees of freedom (Diggle et al.,
1994; Drapper and Smith, 1998). In each test, if the $p$-value did not exceed $0.05$ then the predictor was provisionally retained,
otherwise it was dropped, and the next predictor was considered. When all the predictors had been considered, the $p$-values
for the Wald test on each were compared to thresholds according to the $\alpha$-wealth controlling the false discovery rate. The
successive hypothesis were tested in the order (1) annual precipitation, (2) annual temperature, (3) slope, (4) topographic
index, (5) enhanced vegetation index, (6) MODIS band 7, (7) MODIS band 2, (8) MODIS band 1 and (9) MODIS band
3 (Table 2). The predictors whose $p$-values were below the thresholds would be used in the final model. The models were
sequentially fitted starting with a 'null' hypothesis with the linear spatial trend identified in the exploratory analysis. The were
fitted with robust REML with $c = 2$, in order to avoid problems of convergence. Convergence problems often depend on the
data and occur when $c$ is low and a numerical solution of the equations would not be found (Papritz, 2021). This occurs when
there is low spatial correlation (near pure nugget variogram) and/or poor linear relationship between response and covariates.
However, the final model was fitted with a much lower tuning parameter, $c = 1.75$, in order to penalise the outliers in $\mathrm{Se_{sol}}$
dataset with lower weights.

### 4.3 Quantile random forest algorithm for spatial prediction

A random forest is an ensemble of tree-structured predictors formed by a collection of classification and regression trees
(CART), that depend on the value of a random vector sampled independently, with the same distribution for all the trees in the
forest (Breiman, 2001). It is a method often used for performing predictive tasks (e.g., Nussbaum et al., 2018) by combining
large number of regression trees by the mean of their predictions. Decision trees are often referred to as a 'greedy algorithm',
because each split reduces the residual sum of squares for that particular subgroup, rather than optimising splits to reduce
overall residual sum of squares (James et al., 2013). This 'greedy' property tends to over-fit the training data and results in
poor predictions (Bramer, 2020). Due to their hierarchical nature decision trees tend to be unstable and have large variance
(Hastie et al., 2009), in the sense that large changes in model and prediction following only small changes in the training data.
On the other hand, on average tree based predictions tend to be unbiased. Therefore, algorithms like random forest have been
developed to balance for the instability of CART, but to still be able to profit from the complexity of interaction-type response-
covariate relationships. Quantile random forest is an expansion of random forest that allows for uncertainty quantification for
each prediction (Breiman, 2001; Meinshausen, 2006). We used the Boruta algorithm that uses a wrapper approach for variable
selection using the Boruta package (Kursa and Rudnicki, 2010). The algorithm creates a shadow attribute dataset consisting of
randomly shuffled predictors. A random forest model is fitted including original and shuffled predictors and variable importance
is computed. Variables that have on average larger importance than the randomised variables will be used in spatial prediction.
The quantile random forest is explained in greater detail by (Meinshausen, 2006). The ranger function of the ranger package
(Wright and Ziegler, 2017) was used for fitting the random forest to predict $\mathrm{Se_{sol}}$ in Malawi. In order to directly compare to
robust REML E-BLUP, $\mathrm{Se_{sol}}$ was also transformed by natural logarithm scale.



## 4.4 Measures of uncertainty

Prior and posterior measures of uncertainty can be used to quantify uncertainty in spatial predictions. A prior measure of uncertainty results from the prediction process. Kriging variances are example of prior measure of uncertainty for LMM. The kriging variances are the expected square difference between predicted and the observed values and they can be plotted on a map to show areas where additional sampling is required to reduce this uncertainty. The appropriateness of the kriging variances can be assessed by the standardised squared prediction error (SSPE) after internal cross-validation, with leave-one-out or $K$-fold cross-validation. We used cv.georob function to perform $K$-fold cross-validation for the robust REML E-BLUP. The model is re-fitted 10 times by robust REML but each time $1/K$th of the data is excluded (Papritz, 2021). After cross-validation, the SSPE is computed by

$$\theta(\mathbf{x}) = \frac{\{z(\mathbf{x}_i) - \tilde{Z}(\mathbf{x}_0)\}^2}{\hat{\sigma}_{\mathrm{K}}^2(\mathbf{x}_0)}, \tag{2}$$

where $\tilde{Z}(\mathbf{x}_0)$ is the kriging prediction of $z(\mathbf{x}_i)$ and $\hat{\sigma}_{\mathrm{K}}^2(\mathbf{x}_0)$ is the kriging variance. The expected value of $\theta(\mathbf{x})$ is 1, this is not a sensitive diagnostic. Assuming that the errors follow a Gaussian distribution $\theta(\mathbf{x})$ is expected to have a $\chi^2$ distribution with one degree of freedom, so that the median value of $\theta(\mathbf{x})$ over all data can be used as a diagnostic (Lark, 2000).

The prior measures of uncertainty for random forests are the out-of-bag mean square error and the quantile regression forest that estimate conditional distribution of the predicted variable. About a third of the samples in the random forest are left out during the bootstrapping of samples. The out-of-bag serve as test sample to assess the prediction accuracy of random forests through computation of universal measures of uncertainty (e.g., mean square error) as with cross-validation. Prediction intervals can be computed from conditional quantiles by using quantile regression forest, a generalisation of random forests by Meinshausen (2006).

The prior measures of uncertainty, kriging variances and conditional quantiles, can be used to compute conditional probabilities that–given the current model– future observations of $\mathrm{Se}_{\mathrm{sol}}$ fall bellow a threshold, $\mathrm{Se}_{\mathrm{threshold}}$ (1.49 µg kg$^{-1}$). With robust REML E-BLUP, an assumption of normality of the prediction errors, after cross-validation, should be plausible. Exploratory analysis of the kriging errors after computing $K$-fold cross-validation, showed that the errors could be regarded as a normal random variable. Conditional probabilities also can be obtained from quantile predictions of quantile regression forest without assumption of normality as in the geostatistical approach. Conditional probabilities for a true value exceeding a threshold, $\mathrm{Se}_{\mathrm{threshold}}$, are taken from the full predictive distribution resulting from quantile regression forest (Meinshausen, 2006).

We also used indicator kriging to obtain the conditional probability. The kriging predictions were made by ordinary kriging of a transformed variable, the indicator variable, $\omega$ (Webster and Oliver, 2007). The transformation is made by:

$$\omega = \begin{cases} 1 & \text{If } \mathrm{z}(\mathbf{x}) \le \mathrm{z}_{\mathrm{t}}, \\ 0 & \text{otherwise.} \end{cases} \tag{3}$$

Posterior measures of uncertainty depend on the primary data and are obtained from validation. Validation compares the prediction at a site and the measured value at that site. Several approaches can be used to validate predictions and these include





Jack-knifing/ data splitting and collection of an independent dataset. To assess random forest, OK with GA and robust REML E-BLUP, we validated the predictions with an independent dataset. We computed the universal prediction accuracy measures (e.g., mean error, mean square error, root mean square error). We also computed the mean squared error skill score (MSESS)

$$\text{MSESS} = 1 - \frac{\sum_{i=1}^{n} \left( \{z(\mathbf{x}_i) - \tilde{Z}(\mathbf{x_0})\} \right)^2}{\sum_{i=1}^{n} \left( z(\mathbf{x}_i) - \frac{1}{n}\sum_{i=1}^{n} z(\mathbf{x}_i) \right)^2}, \tag{4}$$

where $\tilde{Z}(\mathbf{x}_0)$ is the prediction and $z(\mathbf{x}_i)$ is the measured value of $\text{Se}_{\text{sol}}$. The MSESS can be interpreted in a number of ways. At the first instance a score of 1 indicates perfect predictions and the root mean square error would be 0. Second, a score of 0 shows that the predictions have the same variance as the data of the validation set and a score less than 0, suggests that the predictions have larger variance than the validation dataset (Wilks, 2011).

Coverage probabilities were estimated from 0.5 to 0.99 for the predictions from cross-validation and of the independent dataset for OK with GA, robust REML E-BLUP and random forest. We used the blakerci function of the PropCIs package for the R platform to compute the 95% confidence interval for each estimated coverage (Blaker, 2000). Chagumaira et al. (2022) showed that a critical probability, $P_t$, can be elicited from a diverse group of stakeholders (S1 to S3) when provided with maps conditional probabilities of not exceeding a threshold. The $P_t$ is an indication of a stakeholder's judgement when making a decision for an intervention using uncertain spatial information. The elicited $P_t$ can be used to assess the validity of coverage probabilities of prediction intervals for probabilities close to a threshold. In this case study we used a range of notional $P_t$ to assess coverage probabilities of prediction intervals and thereby show proportion of mapped area under the selected critical probability value. This analysis can allow stakeholders to identify locations for setting up field trials.

## 5 Results

### 5.1 Linear mixed models for spatial prediction

After transforming $\text{Se}_{\text{sol}}$ by GA, there was no evidence of a spatial trend. Summary statistics for the cross-validation errors for OK with GA are presented in Table 1. The variograms for the transformed $\text{Se}_{\text{sol}}$ are shown in Figure A1. After cross-validation, the variogram estimated using the Matheron estimator has the largest value of median of SSPE of 0.39. The median unbiased back transformed spatial predictions of the concentration $\text{Se}_{\text{sol}}$ in Malawi, by ordinary kriging, are shown in Figure 3a. The map shows that there is higher concentrations of Se in central, northern and southern west parts of Malawi.

After sequential fitting of the models, with robust REML, with ordered predictors for $\text{Se}_{\text{sol}}$ shown in Table 2, enhanced vegetation index (EVI) was selected through the false discovery rate control procedure (Figure 4). The graph (a) shows $\alpha$-wealth over the sequence of tests and the lower (b) shows the $p$-values and the corresponding thresholds under the false discovery rate control with the $\alpha$-investment. The final model fitted with robust REML, with $c = 1.75$, was used for spatial prediction of $\text{Se}_{\text{sol}}$. The maximum likelihood variogram for $\text{Se}_{\text{sol}}$ is presented in Figure A2. The variogram shows strong spatial autocorrelation in the data for $\text{Se}_{\text{sol}}$.



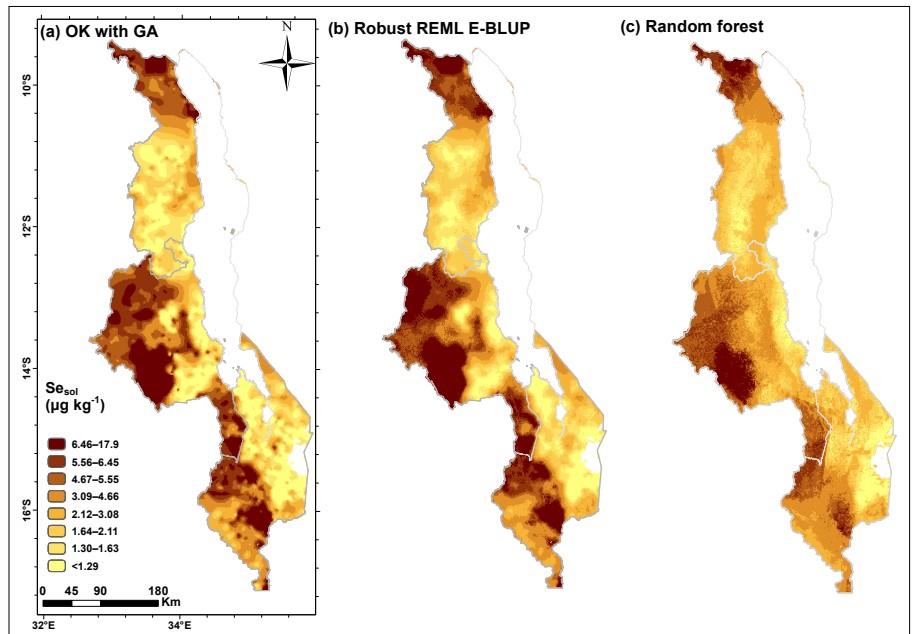

**Figure 3.** Spatial predictions of $Se_{sol}$ concentration across study area by: (a) OK with transformed $Se_{sol}$ (b) robust REML E-BLUP and (c) random forest.

**Table 3.** Models for $Se_{sol}$ concentration in Malawi fitted using robust REML.

| Predictand | Coefficient | | | | $\breve{R}^2_{adj}$ | $\tau^2$ | $\sigma^2$ | $\phi$ |
|---|---|---|---|---|---|---|---|---|
| | $\beta_0$ | $\beta_1$ | $\beta_2$ | $\beta_3$ | | | | |
| | | Easting | Northing | EVI | | | | |
| Null model | 11.0377 | -0.0042 | -0.0009 | | | 0.1095 | 0.4087 | 28.0000 |
| +EVI | 11.7980 | -0.0045 | -0.0010 | 0.0002 | 0.025 | 0.1099 | 0.3985 | 28.0000 |
| Final model | 11.7847 | -0.0045 | -0.0010 | 0.0002 | | 0.1065 | 0.4026 | 28.0000 |

† The symbols $\beta_0$ to $\beta_2$ are the fixed effects coefficients, $\beta_0$ is a constant and $\beta_i$ is the coefficient for the $i$th random effect; $\kappa$ is the smoothness parameter of the correlation function; $\tau^2$ is the nugget variance; $\sigma^2$ is variance of the correlated random effect; and $\phi$ is the distance parameter. $\breve{R}^2_{adj}$ is the difference between $\sigma^2$ of the null model and proposed model expressed as a proportion of the variance for the null model.

The variance parameters estimated by robust REML E-BLUP for the null model and the model with EVI as a predictor are shown in Table 3. A small proportion of the spatially correlated variation ($\breve{R}^2_{adj} = 0.025$) is accounted for by inclusion of EVI as a predictor of $Se_{sol}$.

Summary statistics for the cross-validation errors are shown in Table 1. After cross-validation we computed SSPE, and the median value of $\theta(\mathbf{x})$ was 0.427. This value lies within the 95% confidence interval for the expected value of median under a





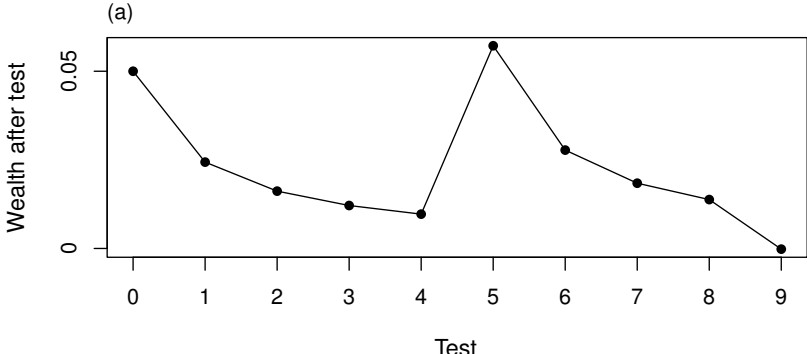

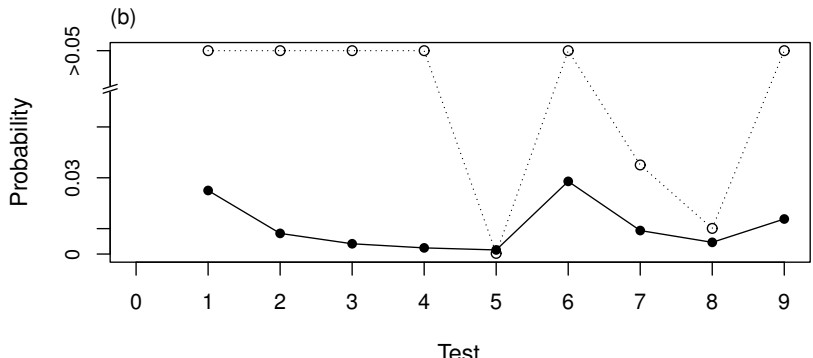

**Figure 4.** Ordered tests for covariate selection for models for $Se_{sol}$ fitted by robust REML. The sequence of predictors is as given in Table 1b. The graph (a) shows $\alpha$-wealth over the sequence of tests and the lower (b) shows the $p$-values for successive tests (open symbols) and the corresponding threshold values with marginal false discovery rate control.

valid model given the number of observations. The mean unbiased spatial predictions of the concentration $Se_{sol}$ in Malawi, by robust REML E-BLUP, are shown in Figure 3b.

Figure 5 shows the maps of conditional probability of $Se_{sol}$ falling bellow $Se_{threshold}$ (1.49 µg kg$^{-1}$). Figures 5a to d are

the probabilities expressed on a numerical scale, and 5e to f shows the same conditional probabilities with calibrated phrases on the IPCC scale. Figure 5a and e correspond to OK with GA. The Figure 5b and f are for robust REML E-BLUP. Figures 5c and g correspond to the indicator kriging predictions. The maps quantifying the uncertainty would be used with the spatial predictions by the sponsors (S2) and information user (S1) to identify sites where trials can be established, locations with $Se_{sol}$ < 1.49 µg kg$^{-1}$.



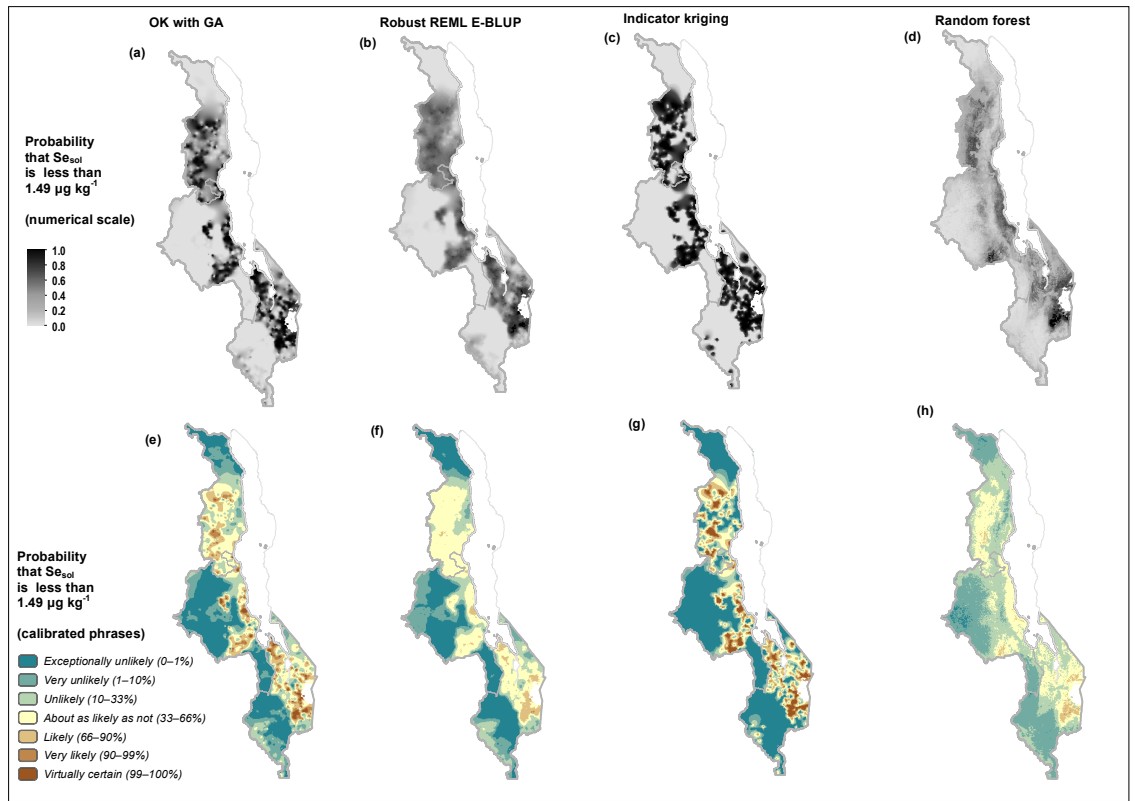

**Figure 5.** Probability that soil Se concentration does not exceed $Se_{threshold}$, 1.49 µg kg$^{-1}$, expressed on a numerical scale (a to d) and according to calibrated phrases (e to h) for spatial predictions by OK with GA (a & e), robust REML E-BLUP (b & f), indicator kriging (e & g) and random forest (d & h).

## 5.2 Quantile random forest algorithm for spatial prediction

All the available covariates (see Table S1) and the spatial coordinates were used to fit a random forest model. Figure 6 shows the box plots of variable importance of original predictors (green) compared to the minimum, mean and maximum importance of randomly shuffled shadow predictors (dark blue) as computed by the Boruta algorithm from 100 repetitions. No covariate was deemed unimportant because all the variables have larger importance than the randomly shuffled shadow predictors (dark blue). The results also showed that downscaled mean annual precipitation and spatial coordinates were the most important covariates. The spatial coordinates were among the three most important covariates for the random forest algorithm, and this reflected the strong spatial autocorrelation shown the variogram for $Se_{sol}$ (Figure A2). Therefore, spatial structure in random forest was modelled splitting the area based on north-south and east-west directions.

Summary statistics for the out-of-bag cross-validation are presented in Table 1. Table 4 shows the parameters for the random forest algorithm, the out-of-bag MSE and $R^2$ were 0.25 ands 63.5%, respectively. The spatial predictions of the random forest





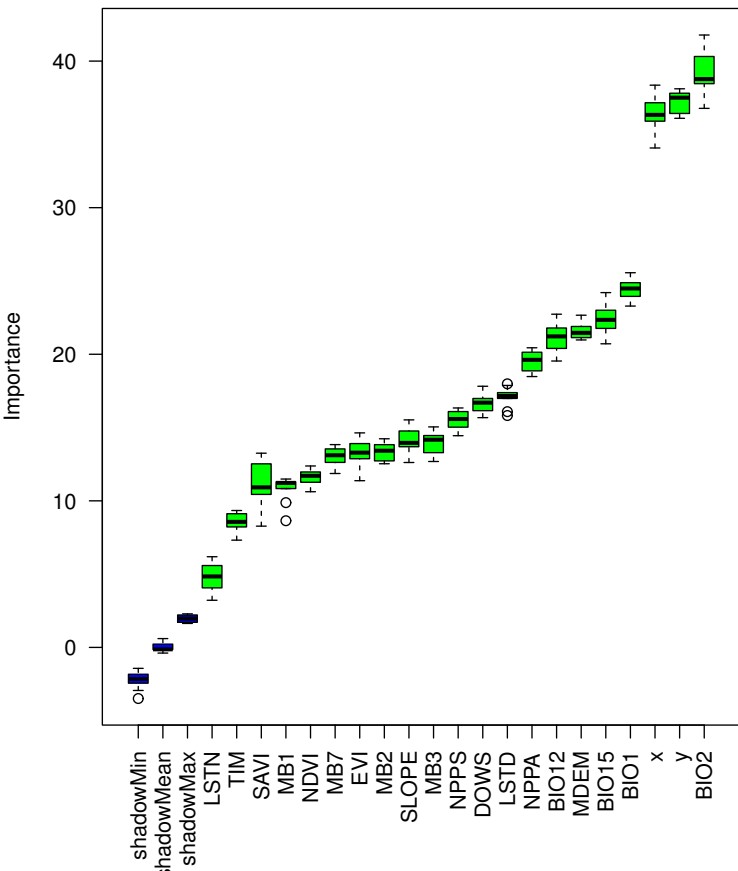

**Figure 6.** Boxplots of variable importance of original predictors (green) compared to the minimum, mean and maximum importance of randomly shuffled shadow predictors (dark blue) as computed by the Boruta algorithm from 100 repetitions.

**Table 4.** Parameters of the random forest algorithm for prediction of $Se_{sol}$.

| Number of trees | Predictors | $m_{try}$† | Out-of-Bag MSE | Out-of-Bag $R^2$ |
|---|---|---|---|---|
| 1000 | 21 | 4 | 0.253 | 0.635 |

$m_{try}$† number of randomly chosen variables.

are shown in Figure 3c. This map is similar to those produced by the LMMs, there are higher concentrations of Se in central, northern and southern west parts of Malawi. The conditional probability of $Se_{sol}$ less than $1.49\,\mu g\,kg^{-1}$, expressed on numerical scale and according to calibrated phrases are presented in Figure 5d and h, respectively.





### 5.3 How can stakeholder (CS) compare predictions?

The posterior measures of accuracy and precision for spatial prediction, for OK with GA, robust REML E-BLUP and random forest, computed after validation with an independent dataset are presented in Table 5. The mean error for OK is $0.07$ on the GA transformed scale and this is close to zero. The mean error for robust REML E-BLUP and quantile random forest are $-0.050$ and $-0.049$ log ($\mu g\ kg^{-1}$), respectively. The mean error for both robust REML E-BLUP and random forest are close to zero. Robust REML produces more accurate predictions of the conditional mean than random forest. The MSESS is the

only statistic comparable amongst the three spatial prediction methods. The score for the OK with GA is $1.04$, and for robust REML E-BLUP and random forest the scores are $0.94$ and $0.87$, respectively. All the models for spatial prediction performed well.

**Table 5.** Accuracy and precision of the predictions computed with an independent validation dataset (n=160).

| Method | Measure | | | |
|---|---|---|---|---|
| | ME | MSE | RMSE | MSESS |
| Ordinary Kriging | 0.07 | 0.400 | 0.448 | 1.04 |
| robust REML E-BLUP | $-0.050$ | 0.212 | 0.326 | 0.94 |
| Quantile random forest | $-0.049$ | 0.216 | 0.342 | 0.87 |

ME–mean error; MSE–mean square error; RMSE–root mean square error, MSESS–mean square error skill score. The units for ME, MSE and RMSE are different. We can only compare the predictions using MSESS.

Figure 7 shows the coverage probability plots for the OK with GA, robust REML E-BLUP and random forest. Figure 7a to c, shows the coverage probabilities with 95% confidence interval for the cross-validations of OK with GA, robust REML

E-BLUP and random forest predictions. All show some deviation from the bisector, with the largest difference seen for the robust REML E-BLUP predictions. Figures 7d to f, show the corresponding coverage probabilities for the validation sites (n=160). The closest agreement is for OK with GA predictions, with the 95% confidence interval included the bisector for all probabilities less than 0.9. The largest deviation is seen for the robust REML E-BLUP. The coverage is less than the specified probability showing that the prediction intervals are to conservative. This may reflect a bias introduced by the tuning parameter

or the consistency correction.

Figure 8a shows the proportion of points at which $P\left(Se_{sol} < 1.49\right)$ exceeds different values of $P_t$ based on cross-validation, and (b) shows the proportion of those sites at which the observed $Se_{sol}$ meets the condition. The robust REML E-BLUP has the smallest proportion of mapped area at which the $P\left(Se_{sol} < 1.49\right)$ exceeds different values of $P_t$. Ordinary kriging has the largest proportion, whilst the proportion of random forest and indicator kriging are similar. Figure 9 shows the proportion of

points based on validation with an independent dataset. Note that for both the cross-validation and independent validation the number of points for which the estimated $P\left(Se_{sol} < 1.49\right)$ exceeds large values goes to zero or small values, so the proportion of the points for which $P\left(Se_{sol} < 1.49\right)$ may vary strongly, or the be undefined. The random forest have the smallest proportion





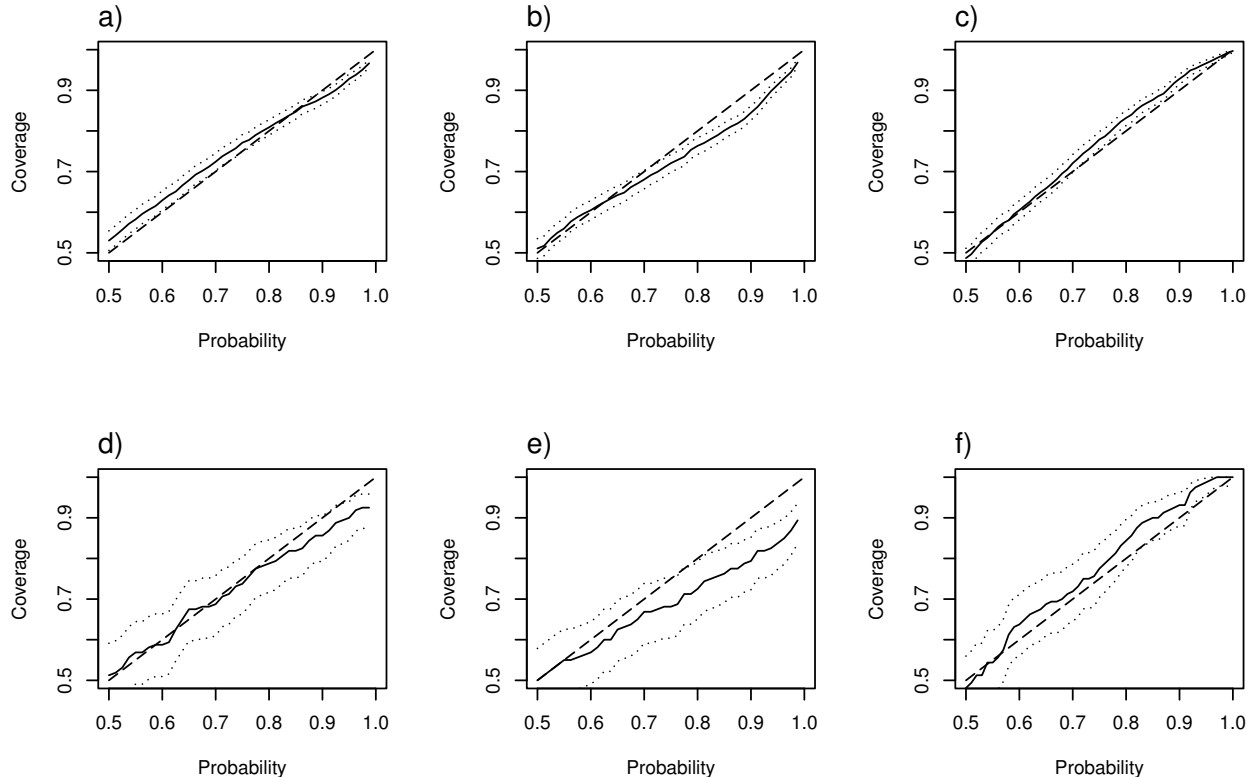

**Figure 7.** Plot of estimated coverage of prediction intervals from 0.5 to 0.99 at the validation locations with their 95% confidence intervals (dotted lines), for: (a) ordinary kriging after leave-one-out cross-validation, (b) robust REML E-BLUP after $K$-fold cross-validation, (c) random forest after validation with out-of-bag data, (d) ordinary kriging after validation with an independent dataset, (e) robust REML E-BLUP after validation with an independent dataset, and (f) random forest after validation with an independent dataset.

of points at which $P\left(Se_{sol} < 1.49\right)$ exceeds different values of $P_t$. The proportion of points for indicator kriging, robust REML E-BLUP and ordinary kriging are similar.

# 6 Discussion

## 6.1 Can we meet requirements of the soil information user?

In this study we postulate a set of stakeholders who require soil information to make a decision on were to locate agronomic biofortification trials to address human Se deficiency. Our objective is to appraise three common approaches to DSM in the light of a decision framework for such stakeholders.




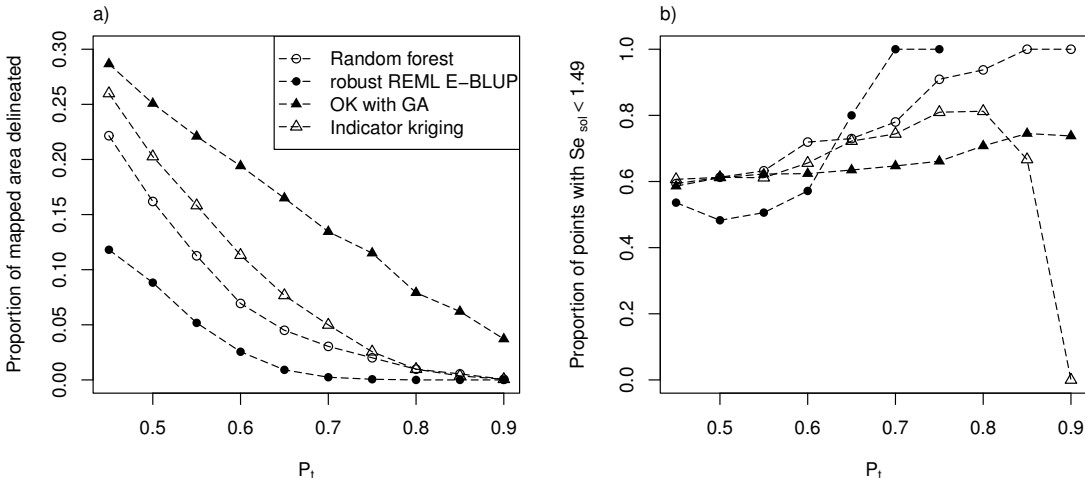

**Figure 8.** Plots for (a) the proportion of points at which $\mathrm{P}\left(\mathrm{Se}_{\mathrm{sol}} < 1.49\right)$ exceeds different values of $P_t$ based on cross-validation, and (b) shows the proportion of those sites at which the observed soil Se meets the condition.

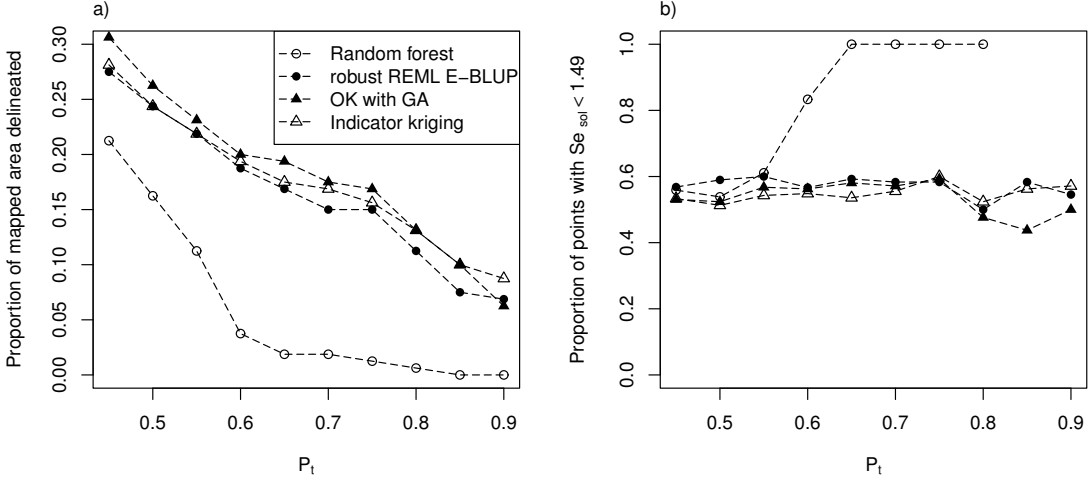

**Figure 9.** Plots for (a) the proportion of points at which $\mathrm{P}\left(\mathrm{Se}_{\mathrm{sol}} < 1.49\right)$ exceeds different values of $P_t$ based on validation with an independent dataset (n=160), and (b) shows the proportion of those sites at which the observed soil Se meets the condition with independent dataset.

The first observation is that the predictions, as measured by an independent validation set, are more precise for the robust REML E-BLUP predictions from the linear mixed model (smaller mean square error) and have a smaller bias. These differences





are not large. However, it is notable that this smaller prediction error variance is achieved by the robust REML E-BLUP with a smaller final covariate set than was used by the random forest, and a model of spatial dependence for the unexplained variation in the target variables.

Second, we note that there is substantial uncertainty in the spatial predictions of $Se_{sol}$ by all three methods. Even though the maps of $Se_{sol}$ show clear structure, which may encourage the user to assume that it has succeeded in representing variation of this property, a local prediction may have substantial error. That is why the uncertainty of these predictions should be explicitly quantified and presented a way that is accessible to the stakeholders (S1, S2 and S3). The objective is that the stakeholder understands the uncertainty and is equipped to use the information with appropriate caution, and attention to the possible

outcomes, when deciding where to locate the agronomic trials to minimise losses. The key challenge is how to characterise the sources of errors with probability distributions (Heuvelink, 2018) and attention should be paid to the assumptions made when modelling uncertainty (Szatmári and Pásztor, 2019).

Geostatistical models and ML methods permit a sophisticated and robust quantification of the uncertainty in spatial information but communicating uncertainty is a challenge. Communicating uncertainty depends on the subject matter and knowledge

of the target audience (Milne et al., 2015). Uncertainty in spatial predictions can be quantified by either using general measures (prediction error variances and prediction intervals) or methods based on interpreting probabilities based on exceeding a threshold. Prediction intervals are commonly used in DSM to quantify and communicate the uncertainty of spatial predictions, and this has been applied in many Pedometrics studies. Chagumaira et al. (2021) found that the diverse group of stakeholders (S1,S2, and S3), find methods of communicating uncertainty based on specific interpretation of the uncertainty to be clearer

and easier to interpret (e.g.,- the probability that the concentration of a micronutrient in grain does not exceed a nutritionally-significant threshold) than general measures such as prediction intervals. Chagumaira et al. (2022) showed that a further step, from just computing conditional probabilities, should be taken to consider how uncertainty can be communicated to a range of end-users for decision-making.

In our case study we have shown conditional probabilities of not exceeding a threshold can be computed for the spatial

predictive methods we used. These conditional probabilities will be used by stakeholders (S1 to S3) to make decisions where to establish the trials. Using the decision process, stakeholders will be provided with information about $Se_{sol}$ concentration not exceeding a hypothetical threshold, $1.49\,\mu g\,kg^{-1}$, to locate sites with inadequate Se supply (I3) in the soil at farm scale (I2). A community would be listed for a trial because the predicted $Se_{sol}$ concentration falls below a threshold. Given the uncertainties in the spatial predictions, it might be possible to set up trials at a location where there is sufficient Se supply. Sites for a trial

should only be set up if the prediction is found to be correct with respect to the threshold. Stakeholders should be assisted when making this decision. Chagumaira et al. (2022) showed that a critical probability value ($P_t$), at which a stakeholder would judge an intervention to address MND deficiencies, can be elicited from a diverse group. The $P_t$ at which stakeholders would judge for an intervention reflects the stakeholders judgement of the losses under different outcomes.

The use of $P_t$ to illustrate different proportion of mapped area at which $P\,(Se_{sol} < 1.49)$ exceeds different values of $P_t$,

where trials may be established, has been presented in Figures 8 and 9. Stakeholders can set a critical value of their choice considering their different circumstances and information needs. For example, a government research organisation with good




and well established linkages with communities, confident that it can engage with minimum disruption if a local sample shows $Se_{sol} > 1.49$, can set $P_t$ to $0.6$. On the other hand a non-governmental organisation beginning local engagement with a community, that maybe more nervous about project withdrawal, may set $P_t$ to $0.8$ and will have a smaller proportion of the
mapped area where the concentration of $Se_{sol}$ meets the condition of $Se_{sol} < 1.49$.

## 6.2    How the spatial predictive methods performed?

In our example, we considered the spatial prediction of $Se_{sol}$ with OK, robust REML E-BLUP and quantile random forest. Ordinary kriging and robust REML E-BLUP are models for spatial prediction that capture the spatial dependence of soil variation (Webster, 2000; Webster and Oliver, 2007). Whereas, random forest are non-spatial and do not capture spatial dependence of
soil variation (Heuvelink and Webster, 2022), although this is the case spatial dependence will be implicitly captured through the input variables. We expected similar results under these models.

In our case study, $Se_{sol}$ was transformed by natural logarithms to make the assumption of normality plausible for robust REML E-BLUP. The log transformed $Se_{sol}$ variable was used also for spatial prediction with random forest. After the transformation, the data showed traits of being bimodal (see Figure S3). We took a further step to use more robust transformation,
Gaussian anamorphosis. The transformed $Se_{sol}$ (see Figure S2) was used for spatial prediction with OK.

The cross-validation coverage probabilities are shown in Figure 7a, b and c. Our results shows that after cross-validation of the OK model indicated it to be a good model and produced accurate predictions of the conditional mean. The coverage probabilities of OK, however were above and below the nominal coverage at lower and higher probabilities, respectively but with quite small deviance. The coverage probabilities for robust REML nearly follow the nominal coverage at lower
probabilities only and then deviate the bisector at larger probabilities. Robust REML E-BLUP produced accurate predictions of the conditional mean and seems to underestimate the error. The random forest, have much better prediction intervals when compared robust REML E-BLUP predictions. The coverage probabilities for random forest deviates from the nominal coverage at higher probabilities. Therefore, the random forest overestimates the uncertainty. Spatial predictions of random forest are less extreme and are central to the mean of the distribution. However, random forest has the larger error when compared to robust
REML E-BLUP (Table 5).

The coverage probabilities for the independent dataset was wide and nearly followed the bisector line for OK and random forest. However, OK with GA and random forest deviate from the bisector in a opposite directions at larger probabilities. Random forest has uncertainty is better quantified, so the tendency to shrinkage in the prediction is not necessarily a strength. The mapped validations also show that OK with GA is arguably better. The validation with independent dataset also confirm
the underestimation of uncertainty by robust REML E-BLUP. The underestimation of uncertainty may be due to the fact we had to use robust methods, which may have down weighted observations in the tails too strongly, such that we do not see the evidence for over-conservative uncertainty quantification by random forest.




## 6.3 Way forward

This study aimed to address challenges of the future of pedometrics (Wadoux et al., 2021) by analysing a decision process

that can be used when mapping micronutrients at a national-scale in sub-Saharan Africa region. In study we aimed to how key users of soil information can be incorporated when designing surveys, mapping, and quantifying uncertainty of the spatial predictions. Many DSM studies put emphasis of comparing performance of machine learning algorithms and statistical methods of spatial prediction (e.g., Vaysse and Lagacherie, 2017; Szatmári and Pásztor, 2019; Makungwe et al., 2021). It is not enough to only quantify uncertainty and leave it there. This is of little use for key users of soil information who have to make decisions

at farm-level, field-scale, regional level, national-level and at policy level. Lark et al. (2022) suggested the need of paying attention to 'decision-focused' measures of uncertainty regardlessly of the method of spatial prediction used. Further steps such as investigating how stakeholders use probabilistic representation of uncertainty is of paramount importance. Chagumaira et al. (2022) showed how discussions centred on how probabilistic representation of uncertain information, with diverse stakeholders can be used to elicit critical probabilities at which they would recommend an intervention.

In the GeoNutrition project, we aim at using spatial information to target areas where specific interventions would be appropriate to efficiently use the scarce financial resources. This is important because most people in countries south of the Sahara (e.g., Ethiopia, Malawi, Zambia, and Zimbabwe) mainly rely on subsistence farming for their food and income. The decision process presented in this paper would be important in addressing some questions raised when addressing micronutrient deficiencies. This decision process may be applied for a different problem, e.g., decisions on sampling, and thorough decision

analysis is required when addressing such problems.

## 7 Conclusions

In this study we presented a framework of decision process that can be used when mapping micronutrient at a national-scale in sub-Saharan Africa region and this is a significant step towards addressing the challenges of the future of pedometrics. We also showed how key users of soil information can be incorporated when designing surveys, mapping and quantifying uncertainty

of the spatial predictions. We have shown how the decision-process for making decisions when using different methods for spatial prediction. The linear mixed models (ordinary kriging and robust REML E-BLUP) underestimate the uncertainty in the spatial predictions of $Se_{sol}$, whereas the random forest overestimate the uncertainty. However, the decision to which method is better in providing soil information remains difficult. This study has shown the importance of cross-validation and validation of conditional probabilities used when quantifying uncertainty in spatial predictions using a critical probability threshold. This

allows stakeholders S1 and S2 to make rationale decisions based on their different circumstances and information needs.

## Appendix A

The variance parameters for $Se_{sol}$ transformed by were estimated from fitting an exponential variogram on an empirical variogram estimated by the method-of-moments (Figure A1, Matheron, 1965). The maximum likelihood variogram functions for



the null model (coordinates filtering spatial trend) for $Se_{sol}$ concentration, and for enhanced vegetation index (EVI) added as a
predictor are shown in Figure A2.

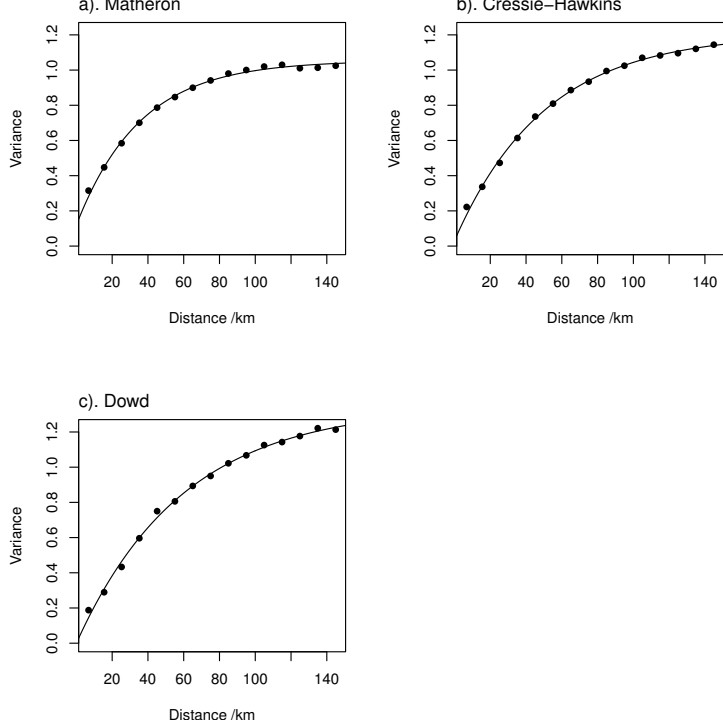

**Figure A1.** Variogram functions for $Se_{sol}$ transformed by Gaussian anamorphosis estimated by (a) Matheron (1962) (b) Cressie and Hawkins
(1980), and (c) Dowd (1984) estimators

*Data availability.*

The data used in this study are available at https://doi.org/10.6084/m9.figshare.15911973.v1 (Kumssa et al., 2022)

*Author contributions.* The project was conceived, designed and implemented by CC, RML, AEM and MN. RML, PCN and MRB were
responsible for project administration and funding. PCN and JGC supervised the data collection. All authors contributed to the preparation
of the article.

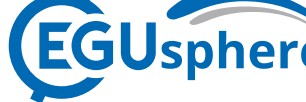



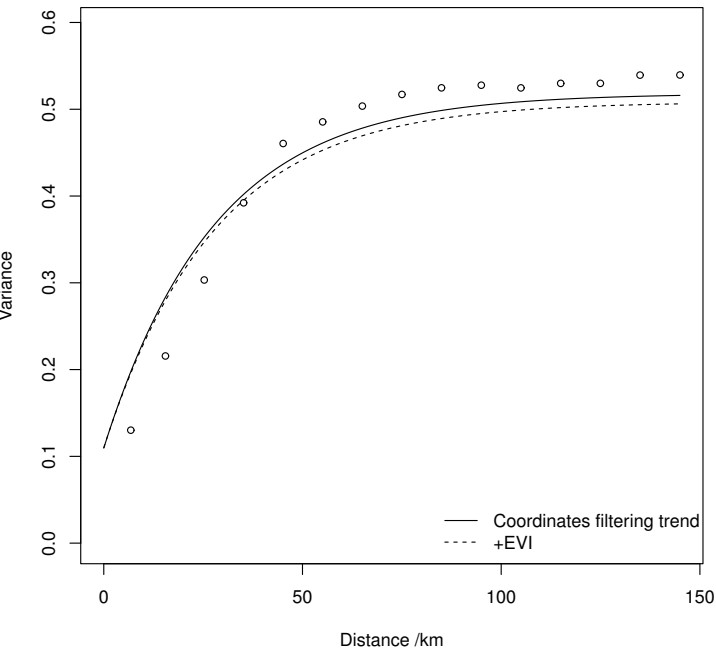

**Figure A2.** Variogram functions for the null model (coordinates filtering spatial trend) for $Se_{sol}$ concentration, and for successive model with selected environmental covariates added as predictors.

*Competing interests.* The authors declare that they have no conflict of interest.

*Disclaimer.* The funders of this paper were not involved in the study design or the collection, management, analysis and interpretation of the data, the writing of the report or the decision to submit the report for publication.

*Acknowledgements.* This work was supported, in part, by the Bill & Melinda Gates Foundation [INV-009129, INV-002855]. This work was
also supported by the Nottingham-Rothamsted Future Food Beacon Studentships in International Agricultural Development. The authors gratefully acknowledge the contributions made to this research by the participating farmers and field sampling teams from the Department of Agricultural Research Services, and Lilongwe University of Agriculture and Natural Resources.



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
