# Peer review of "Mapping soil micronutrient concentration at national-scale: an illustration of a decision process framework"

_EGUsphere, 2022_

## Author Comment (AC2)

**Referee Comments for SOIL manuscript number EGUsphere-2022-583**

Thank you for the opportunity to revise our manuscript. We would like to thank the referee for their time and for the constructive comments they have provided.

**Main comment.** We accept the reviewer's opinion that the first version of the paper did not do enough to illustrate the role of the decision framework. For this reason, we propose to move most of the technical description of the geostatistical and machine learning (ML) mapping (which is not new) to supplementary material, and to address four hypothetical case studies. The first two of these (Case Studies I and II) are concerned with secondary analysis for mapping of the GeoNutrition data on soil Se concentration ($Se_{sol}$). In Case Study I, the stakeholder wants to select sites for agronomic biofortification with feeding studies and is very risk-averse about initiating work with a community where withdrawal might be necessary if the true $Se_{sol}$ exceeds the threshold. In case Study II, the stakeholder is willing to accept a greater level of risk, for reasons we explain. The outputs which address the requirements of these two contrasting end-users of information differ, for reasons which directly reflect their perspectives in ways which the decision framework picks up. In the second pair of case studies (Case Study III and IV) we consider the planning of further sampling for investigation of $Se_{sol}$ in a district of Malawi (an administrative region). In the first, there is to be an intervention planned on the basis of the soil mapping, and the stakeholder wants to limit the risk that sites where the $Se_{sol}$ content is deficient are not mapped as such. In the second the stakeholder is planning general work on Se in soil and crops, and wants a baseline map for reference, requiring only that the procedure seems robust and repeatable. We attach some preliminary outputs for these case studies at the bottom of this document. We believe that these four case studies, and the contrasts between them, illustrate the value of the decision framework for directing digital soil mapping activities in ways which reflect the actual requirements of information users.

We propose to revise the manuscript, incorporating these suggestions. We reply to each of the additional comments below. Our suggested edits in the paper are in blue below, with line numbers indicating where we wish to make the changes.

| Referee Comment | Author response |
|---|---|
| (L27-28) What do the authors mean by 'scales' when they write "inherent variation of soil at multiple scales"? Does 'scale' refer to the extent of the study area, the spatial resolution, or the spatial support? | These interpretations of "scale" which the reviewer offers all relate to the process of measurement or observation. In contrast we are making the common observation that soil variation arises from processes that operate over different characteristic length-scales, for example parent material varies over distances up to km in some environments, while soil variation imposed by bioturbation may impose differences over a few cm. We shall edit the text to read "inherent variation of the soil over contrasting length scales" |
| Authors also mention spatial measurement error as a source of prediction uncertainty, but has this uncertainty source been covered in this study? And what do you mean with "uncertainty arising from predictive factors in our spatial models"? Are these 'predictive factors' the covariates or something else? If these are the covariates, how do covariates lead to uncertainty? Perhaps it is the limited ability of covariates to explain the spatial variation of soil properties, but that is not clearly stated | In this study, we removed the samples that had spatial measurement error as indicated by GPS quality measures, please see L257—259.

We shall clarify that the "predictive factors" are covariates. In any statistical model one source of prediction error, quantified in the standard error of prediction, is the uncertainty with which the fixed effects coefficients are measured. We shall clarify this in revision |
| (L31) Here it writes "organized", while L34 writes "recognise". Decide between UK, US or Oxford spelling and use it consistently throughout the manuscript | Suggested edit on L34 has been made on the manuscript and we have followed H.W. Fowler and Sir Ernest Gowers consistently in the manuscript. |
| (L39) Again, what do you mean by "... soil variation occurs ... at multiple spatial scale in space". Also, sloppy formulation, is not 'multiple spatial scale' by definition something that refers to | Please see our first comment, where we have addressed the issue of scale. "Scale" can also be used in a temporal sense, just as "spatial frequency" is sometimes used in a spatial sense. |

| | |
|---|---|
| space? For instance, there is no such thing as 'multiple spatial scale in time', or is there? | |
| (L45) Confusing text because it suggests that past surveys is the opposite of a systematic grid, as if past surveys could not use systematic designs. | Our point was that a surveyor starting from scratch might choose a systematic grid, but they might also assemble data from past surveys when they cannot choose the design. We think that is quite clear here. |
| (L67-68) This makes no sense. Variable selection methods are also used by ML methods (e.g., Recursive Feature Elimination). Moreover, kriging with external drift also weights covariates 'appropriately'. | We shall edit the text on L67—68

Some ML methods use variable selection whilst others aim to weigh all covariates appropriately. |
| (L77-78) This comes closest to a sentence that states the objective(s) of this paper, but it is not clearly formulated. It would help the reader if the aims were clearly communicated at the end of the Introduction. | This is a fair point, and we shall edit the text on L77— 79 (new final paragraph to section 1).

We shall set out a decision process which can be used to clarify the requirements of end-users of information, sponsors of data collection and other stakeholders. This process will address how uncertainty affects the use of the information and will assist all parties to identify how this uncertainty can usefully be quantified in ways which address the user's requirements and can be effectively communicated to them. We then illustrate this framework with four notional case studies, using data on soluble Se content of soils in Malawi. Two case studies are secondary data analysis, where two contrasting users both require soil information to support a decision on the selection of trial sites but have different attitudes to risk. Two other case studies postulate further sampling within an administrative region, both to support mapping, but for a clearly defined task in one case, and to provide baseline information in a second. The case studies will illustrate how the decision framework can be used to ensure that a digital soil mapping task is not planned or executed in isolation from the end user. This study fits well the research agenda… |

| | |
|---|---|
| (Introduction) The Introduction pays much attention to soil mapping, geostatistics en machine learning for soil mapping, including sampling design optimisation (L23-L68), giving the text a flavour of a review paper, while this is meant to be a research paper. Much less attention is paid to a review of how prediction uncertainty is quantified and communicated (L69-L77), while this is the focus of this paper. I sense a disbalance. | The aim of the introduction is to state our position that (i) geostatistical studies in soil science very rarely start with a consideration of the end user's requirements and response to uncertainty and that (ii) these considerations should be brought in from the start, including at the stage when decisions are made on sampling design.  This is not just a matter of paying more attention to how prediction uncertainty is quantified and communicated, because this too must be informed by a systematic consideration of user requirements. We shall restructure the introduction along the following lines

i.   Soil information is needed by various stakeholders, exemplified from the MND question.

ii.  Geostatistical and ML methods provide a basis for making predictions from data and covariates, with quantification of uncertainty, and provide (less so for ML) a basis for planning sampling.

iii. The key point on the lack of focus on end-user requirements in pedometrical studies, particularly regarding uncertainty (as in the final paragraph of the original paper).

Communicating uncertainty should be considered in the context of specific interpretation of the spatial information (e.g., nutrient concentration relative to a significant threshold) using probabilities can be more accessible to a wider range of end-users (Chagumaira et al. 2021).  That is part, but only part of our decision framework, and we think that the revised paper with greater emphasis on four contrasting case studies will make this clear. |

| | |
|---|---|
| (L84) I am not convinced that they always require reliable evidence. Sometimes it simply is not possible to generate reliable information and so one has to make do with information that is not very accurate. But it may still be better than no information. Our task then is to communicate the uncertainty associated with the information to users and even better explain them how uncertainty can be incorporated in decision making. Is this not what this manuscript is all about? Stating that stakeholders can only work with reliable evidence to act undermines the message of this paper (namely that stakeholders can also make use of uncertain information). | We start from the position that "reliable" does not mean "without uncertainty" because all information in the real world is uncertain. However, we understand how this confusion may arise. We propose to edit the text to read:

These stakeholders require evidence on which to act, in most cases the outcome of their decisions may involve financial costs and implications for farmers' livelihoods, food security and public health. Because, as seen above, information is always more or less uncertain, the stakeholder must be made aware of this, and must be able to make a decision which is as robust as possible in the presence of uncertainty. Pedometrical methods provide a technical basis for this, but the needs of the stakeholder are unsystematically reported, perhaps a lot of consideration of their needs go unaddressed. |
| (L93) Sentence says that there are four questions, but the list has five. | L93 has been edited

This can be characterised in terms of a set, I, of five questions about the information. |
| (L94, I2) This has a link with the spatial support at which information is needed (if a decision is made at farm level, then we need information at farm level, or perhaps not?). Authors will know that the uncertainty associated with soil predictions is strongly affected by the spatial support, but none of this is included in the manuscript. To me this is unacceptable. They should have addressed how uncertainty changes under a change of support and their methods should account for that. | We agree that spatial support is an important point which should be emphasized. We propose to add the following text at the end of 2.1, also expanding on question I5.

Question I5 raises the management unit for which a decision is made. A farmer might make decisions which apply to a field, or to a whole farm, while an administrator might make decisions which apply to a whole district. This may require a change of the support of the soil information (the size and shape of the basic aliquot of soil for which data are obtained) from a point or core |

| | |
|---|---|
| | to a "block" such as a field. This is readily done in the geostatistical setting, for example Gashu et al (2021) present predictions of grain micronutrient content at core support, but also upscaled predictions by block kriging of the mean value across administrative regions (woredas). Such geostatistical predictions on a new support are also provided with model-based prediction error variances which characterise the uncertainty.

Finally, Question I5 may be important when the soil properties to be measured may be of interest beyond the immediate question of the stakeholder. Some soil properties may vary over time in response to management, climate etc. but data on properties such as soil texture may be valuable for future users with different questions, or for those who later want to quantify change in nutrient or organic status. Certain stakeholders, such as Departments of Agriculture, might be willing to incur certain costs in a new survey which are justified by foreseeable future uses of the data in addition to the current problem.

Regarding our case study, the scale of measurement was based on a sub-field sampling support located near the centre of the field to avoid edge effects. As we do not have boundaries for specific fields of interest for predictions, and, at national scale the difference between the subfield and field supports are small, we have not attempted a change of support here. We shall add a sentence to this effect near line L94 |
| (L94, I4) Not clear to me why the "given uncertainties" is included and what it means. Do we get different outcomes from the decisions in case of uncertainties? | "outcome given condition" is normal English usage, but we have tried to make it clearer editing I4 on L94, |

| | |
|---|---|
| | (I4) What set of outcomes from the condition is consistent with the uncertainties in the spatial predictions. |
| (L94, I5) What do you mean by "potential legacy value"? How relevant is this? | See above comment and addition on I5 at the end of section 2.1 |
| (L97) Cryptic sentence, I do not understand what "The state is the state of affairs which our soil information predicts" means. Surely this can be formulated more clearly. | This statement on L97 has been edited as

The state is the condition which our soil information predicts |
| (L103-104) This needs a better explanation. Some farmers may rather apply extra lime if they are uncertain, just to make sure that there is no yield loss. | In the context of smallholder farmers in low- and middle-income countries, whom this study reports, application of lime is very expensive, for example applying 2 tonnes per hectare costs USD$212.72 for lime and labour, this is more expensive than basalt or top-dressing fertiliser (see Lolani and Kabembe 2020). This might look like a paltry figure in developed countries. Furthermore, nutrient lock-up from over-liming is a significant risk and over-liming can cause structural damage to tropical soils due to destabilization of sesquioxide's. Therefore, most farmers will not apply lime where they are uncertain.

We propose to edit the text to emphasize further the risks of over liming, in addition to the economic cost, particularly in a tropical context.

Martin, L., Vernon, H.K., 2020. Potential of reduced agricultural lime application rates to increase yield and profitability of maize through microdosing in central Malawi - A short note. African Journal of Agricultural Research 16, 785–790. doi:10.5897/ajar2019.14619 |

| | |
|---|---|
| (L106-115) I did not find this text very clear. Perhaps including a figure might help? | |
| (Figure 1) I like this figure, although it could perhaps be rearranged a bit so that it is clearer that it is a flowchart from I1 to I4. I also did not understand the "Opportunity cost. Nutrient deficiency" description, did not understand the difference between "Unnecessary cost" and "Opportunity cost" and between "Moderate yield loss" and "Some yield loss". It is also not explained in the main text. | We will edit the figure and put "Unnecessary cost" at both. To distinguish between the yield losses, we will replace "Some yield loss" with "Substantial yield loss." We will add the following text at L108 to explain the unnecessary costs.

The unnecessary costs include purchasing of lime and lime and increased immobilisation of micronutrients. |
| (Section 2.2) The first paragraph gives the impression that authors did not make a thorough literature study and came up with the division in three types themselves. There are no references to that type of literature, but surely there must be lots of literature on defining and classifying stakeholders. I searched for publications in Web of Science that had both the word "soil" and "stakeholder" in the title and got already 35 hits (1752 hits if both terms must appear in the abstract). | There are many typologies for stakeholders, and we are proposing one for our purpose to highlight the stakeholder as a user of information, as the sponsor of information, and the (dys)beneficiary of the resulting decision. This is not a rigid typology because the same stakeholder might perform all roles (e.g., a farmer who pays for soil analysis on their field to make a decision about fertilizer requirement). We developed this terminology to highlight the thinking that is needed about decision making under uncertainty, we have explained it, and we see no reason why we should look for a precedent to justify it. |
| (L129) But there are so many factors other than sample size that influence uncertainty. Why focus on this one only. Why are other factors not mentioned and reviewed? | It is true that many factors contribute to uncertainty in information, but the number of samples and the design is the principal one that we can control, and that determines costs. In some cases, there may also be costs associated with covariates (if these are not open data). These are already highlighted under O6. We propose to add, at L 134. "In addition to sample size, uncertainty of predictions might also be reduced by including suitable covariates in the prediction model, and the costs of |

| | these data, where they are not open, must also be considered. This therefore appears in this set at O6. |
|---|---|
| (L130) Is it really the sponsors and users who make decisions about information? In fact, what do you mean by this? Is it not the surveyor or producer of information that makes decisions about the uncertainty? I would expect that a surveyor informs the user how large the uncertainty will be given a sample size, and how much it will decrease if sample size is increased, so that a user can taken an informed decision about trade-off between costs and accuracy. This is not clearly explained in L130-134. | We do not see the problem here, in fact the reviewer's initial question is addressed in their subsequent remarks that the sponsor/user and producer make this decision jointly, the latter quantifying in some way the link between uncertainty and cost, and the sponsor using that to make a decision on use of resources. In fact, this process is difficult, and in questions O1 – O6 we highlight the main issues. We propose a rewording of the text at lines L129--134 to read:

"The uncertainty of information about the soil arises from variation of the soil. Sample size and distribution is one factor which can be controlled, and a key one determining the cost of information. In addition, predictions might be improved using covariates for modelling, and these may impose an additional cost if they are not from open data sources. Stakeholders therefore need to arrive at a series of decisions which determine quality of the information and the cost of its collection. Expertise relating to the sampling options will usually be limited to the statisticians and other scientists engaged, but the question of the optimal resource investment given the reduction of uncertainty as investment is increased, must be taken by the sponsor in the light of the implications of uncertainty for the data user, and the competing demands for resources. The key questions identified below are framed primarily in technical terms but must be addressed by the scientist in communication with the sponsor and other stakeholders. For example, it is necessary to address O2 and O3 in dialogue with stakeholders (Chagumaira et al., 2021), and then to frame questions O4 – O5 in terms of potential trade-offs between uncertainty and cost. O1 might then be |

| | |
|---|---|
| | addressed by the surveyor when these other questions are resolved". |
| (L135) I wonder how many of these questions are understandable to the stakeholder, I don't think they should be addressed by engaging with stakeholders. These are questions that the surveyor/modeller must answer, the results of which may be shown to stakeholders (by providing them with trade-offs between accuracy and costs). | We address this under the previous point. |
| (L135) What I really miss here is the most important first question of any survey: what is the goal? None of the listed questions can be answered without it. See the excellent book by De Gruijter et al. (2006, https://link.springer.com/article/10.1007/s11004-008-9147-7), which, unlike this paper, takes a very structured and comprehensive approach on how to design a survey while accounting for uncertainty. | See questions I1 – I5. These serve precisely to characterise the goal: what question must be addressed etc. We are very familiar with the book by deGruitjer et al and consider our approach entirely consistent with it. |
| (L137, V1) Of course we can, if we have the data. But do we need it, if decisions are taken at much larger supports (field, farm, district)? | We do not understand what the reviewer is saying here. If we want to link uncertainty to sample effort, for questions O1 et seq. then V1 is essential, even when what we require is a simple estimate of the regional mean. We propose to replace the paragraph which begins at L138 with the following:

An answer to V1 allows us, for a range of strategies, to express how the uncertainty of predictions depends on sample size (O4) and sample distribution (O3). Question V1 might be addressed from a pilot survey, the literature (Paterson et al., 2018) or results from a cognate environment (Alemu et al, 2022). Question V2 is important, because if analytical error is substantial then appropriate duplicate measurements, at additional cost, will be needed. |

| | The questions above allow us to clarify the objectives of a planned survey, the use to which it will be put and the implications of uncertainty (set I). The questions in set O determine a survey strategy, dependent on the background information in set V. The constraint, as introduced above, comes from the cost of field and laboratory costs and other survey activities. The resources questions below are therefore important. |
|---|---|
| | Paterson, S., McBratney, A.B., Minasny, B., Pringle, M.J. (2018). Variograms of Soil Properties for Agricultural and Environmental Applications. In: McBratney, A., Minasny, B., Stockmann, U. (eds) Pedometrics. Progress in Soil Science. Springer, Cham. https://doi.org/10.1007/978-3-319-63439-5_2 |
| | Alemu, R., Gelaw, A.M., Gashu, D., Tafere, K., Mossa, A.W., Bailey, E.H., Masters, W.A., Broadley, M.R., Lark, R.M. Sub-sampling a large physical soil archive for additional analyses to support spatial mapping, a pre-registered experiment in the Southern Nations, Nationalities, and Peoples Region (SNNPR) of Ethiopia. Geoderma, 424, 116013 |
| (L137, V2) Other than what? | See edit above. |
| (L138-139) Data and surveys are costly, but is not a survey a means to get data? So why treat them as two different entities? And of course we should take rational decisions, that is a sine qua non. | See edits in response to comment on line 137 |
| (L142) This does not help me much. What do you mean by an "actual decision on sampling"? What is the difference between | The question R1 is centred on whether a budget for sampling is fixed or not. If it is not fixed then the sponsor has an actual decision on the sampling exercise, rather than simply setting this |

| | |
|---|---|
| R2 and R3? What is the difference between R4 and O6? Covariates for what? | according to the resources available. We propose to clarify this in revision.

R2 and R3—are different. Data are not the same as information. Legacy information might be a set of soil maps or GIS layers. Legacy data would be a set of point observations, or analyses from soil samples. This will be clarified in the revision.

O6 and R4—are very different, Under R4 we consider which covariates are available to us, for example as open data or an affordable cost. Under O6 we consider which covariates are, in principle, ones we should use which might be based on prior experience or expert judgement, as deployed in the case study here through ranking of the covariates |
| (L145) I agree with this but then shed some light on how this is done effectively. Is that not what this paper aims to do? I don't find it addressed and explained. | This paper is not specifically about communication of uncertainty. As the title indicates it presents a framework, in this series of questions, for survey planning, in which uncertainty among other factors, is considered. U1 is then discussed in detail from L150 onward. |
| (L146) What do you mean by "value of uncertain information"? There is a rich literature on "value of information" but that is not mentioned. | There is a substantial literature on the value of uncertain information, the term "value of imperfect information" is perhaps more widely used. We propose to introduce this term at line 150 in the revision.

There is a large literature on question U1, some of which is summarised by Lark et al (2022). Lark et al (2022 and Giasson et al (2000) discuss how Value of Information (VoI) theory (Howard, 1966) can be applied to soil information. The early literature on VoI considered the value of perfect information, but it was realized that an extension to imperfect information (which we can uncertain information) in this paper was possible. The |

general approach is illustrated by Lark and Knights (2015). They considered a loss function, which expresses the costs of making a decision (in their example, about liming the soil) as a function of the error in the information on which that is based (in their example, information on the mean soil pH for a farm based on a simple random sample). The uncertainty of the sample mean is characterised by its standard error, and the loss function can be used to calculate the expected loss from a decision based on an estimate with a specified standard error. The expected loss is reduced as the standard error of the estimate is reduced, but this in turn requires an increased sample size. It is possible to compute the marginal reduction in expected loss as a function of sample size, and to find the sample size at which the marginal cost of reducing the uncertainty equals the marginal reduction in expected loss (Lark et al., 2022). This is the optimal sample size, and the corresponding standard error could be regarded as the acceptable uncertainty in the sample mean, given the costs of sampling and the value of the imperfect information. This would be one answer to U2. U3 is addressed in this example by the fact that the standard error of the sample mean, $\bar{\sigma}$, depends on the standard deviation of the target variable, σ, and the sample size, $n$, for a simple random sample by the familiar relation

$$\bar{\sigma} = \frac{\sigma}{\sqrt{n}}$$

In another example Viscarra Rossell et al (2001) used a loss function for a decision about liming a field to be used for growing wheat (see also Lark et al, 2022). They considered the expected loss at the scale of subfield management units, with the soil lime requirement determined by spatial prediction from sample grids of different intensity. Their conclusion was that there was no

solution at which the value of spatial information justified its cost, indicating that field-scale management of soil pH was most appropriate in that case. These examples illustrate how a consideration of question set U can lead to a practical decision on the collection of soil information.

The process above, based on value of imperfect information, might not always be feasible. For example, the loss function for a particular decision might not be accessible, entailing many intangible considerations such as the political cost associated with making an erroneous decision about environmental management at a site. However, it may be possible to elicit a measure of acceptable uncertainty indirectly. Chagumaira et al (2022) presented stakeholder groups with information on conditions at sites for which a decision was notionally required, with uncertainty expressed in terms of the probability that the value of the soil or crop property of relevance to the decision fell above or below a recognised threshold value on which the decision might be based. It was possible to estimate a probability of this situation at which an intervention would be recommended. As Chagumaira et al (2022) noted, this is equivalent to eliciting how the stakeholder group values the cost of an error of overestimation relative to one of underestimation. Decisions on sampling effort can be supported by calculating the probability that a site where an intervention is required would be identified as such by the data user, as a function of the sample effort (Chagumaira 2022). In some circumstances stakeholders might be able to directly specify an acceptable uncertainty for a predicted value of a variable in terms of its prediction error variance or, equivalently, the standard error of the prediction. For example, Ruffo et al (2005) specified sampling grid spacings for soil nitrogen testing on farms using the relationship between

| | |
|---|---|
| | spacing of a square sample grid and the kriging standard error calculated with the OSSFIM algorithm and specifying an acceptable standard error of the prediction at just over 10% of the upper bound of values proposed as thresholds for intervention.

Ruffo, M. L., Bollero, G. A., Hoeft, R. G., and Bullock, D. G, 2005. Spatial variability of the Illinois soil nitrogen test: implications for soil sampling, Agron. J., 97, 1485–1492,

Viscarri Rossel, R.A., Goovaerts, P., Mcbratney, A.B., 2001. Assessment of the production and economic risks of site-specific liming using geostatistical uncertainty modelling. Environmetrics 12,699–711, doi:10.1002/env.471

Chagumaira, C, 2022. Geospatial modelling of soil geochemistry at national-scale for improved human nutrition. PhD thesis, University of Nottingham. |
| (L147) What do you mean with "can the acceptable uncertainty be quantified? What is the difference between "acceptable uncertainty" and "tolerable uncertainty", how are these defined? Provide references to these terms because I would hope and expect that you build on existing theories. | We will address this point raised by the referee and please see previous comment at L150 in the revision. To explain further, various approaches, c.f. OSSFIM examples, uncertainty acceptable to users as per Chagumaira et al (2022), or the level of uncertainty at which marginal expected loss equals marginal cost of reducing loss via increased sampling as per a full VoI analysis. |
| (L152-157) Difficult to follow without proper explanation. Strange that from a rather general text you move to a very specific and somewhat complicated case. Same applies to
L159-161: what is offset correlation, what is robustness of the final map, what is the final map to start with, why do you need a variogram and why would the offset correlation (or is it the survey | At this point in the paper, we are discussing the decision-making process in general terms, with specific examples to illustrate the different approaches (such as the loss function). |

| | |
|---|---|
| effort?) be sensitive to arbitrary variation of the origin of the survey grid. This is not understandable without proper explanation. If it is made understandable, I wonder how relevant such a very specific case is to the general problem of accounting for uncertainty in decision making. | |
| (L162) We get a case study, but the methodology has not adequately been explained. | The method of providing spatial information needed for this case study were presented from sections 4.1 to 4.5. We will highlight explicitly the link between the two sections. |
| (Figure 2) What is the difference between this figure and Figure 1? What does "Trail established" mean? It is a real pity that this manuscript has so many errors. Why didn't the authors carefully check it before submission, if only out of respect of the reviewers?? | The figure will be edited. |
| (Sections 4 and 5) Now we get the Materials and Methods section, which makes me wonder: what did we get the previous 245 lines? The manuscript is getting very long and I am afraid I am losing it. Perhaps it could have worked if authors had explained the structure of the work at the end of the Introduction, but they didn't. Sections 4.2 and 4.4 are mainly about advanced geostatistical modelling, Section 4.3 on random forest for spatial prediction, so where is the handling of uncertainty in decision making? Where is the link with Section 2? This is only very marginally addressed. The same holds for Section 5, this presents the outcomes of the geostatistical and machine learning analysis, but where is the connection with Section 2? Maybe Section 5.3 aims to do that, but again we get an advanced statistical text with a lot of jargon that has little connection with the | In line with the comments concerns raised by the reviewer, we shall revise the structure of our manuscript. The section 4, on material and methods describing the geostatistical and random forest for spatial prediction will be moved to the Supplement.

To address the concerns of the referee we will add some contrasting case studies. The stakeholders in case study I are very risk-averse and those in case study II are less risk averse. We have described the case study in detail (see Appendix). The stakeholders in case study II—this is another secondary data analysis to map the same variable but with a stakeholders which is already engaged with communities, has a good relationship with them, and is prepared to tolerate a larger risk and their L1/L2 is ~1.2 or so Pt = 0.55. For case studies (Case Study III and IV) we consider the planning of further sampling for investigation of Se in crop and soil in a district of Malawi (an administrative |

| | |
|---|---|
| decision process descriptions addressed in Section 2. Stakeholders will have a hard time understanding this all. I also did not understand why different mapping models have been applied and were compared (OK, REML E-BLUP, RF), Why detailed information about one of these methods (e.g., see Figure 4, but no details about another mapping model (i.e., random forests)? And then in Figure 5 we find that also indicator kriging is included, why not in Figure 3? Section 5.2 then explains the application of a random forest model, but results of this model were already presented in Section 5.1, does this make sense? | region). These case studies will be presented in the revised manuscript. |
| (L483-485) That is what this paper is about and what should have been demonstrated, but that did not happen. | We have revised the manuscript to demonstrated how the decision process can be used by stakeholders when making decision using uncertain information. |
| (L485-487) I do not understand why these statements are made here and what they exactly mean. It is not connected with what was presented before in this section. | We will remove these statements and the discussion will be centred on application of the decision process framework. |
| (L561-570) The Conclusion is poorly written, and many of the claims made here were not realised in this paper. There are quite a few platitudes, and a mishmash of messages. This paper lacks focus and has too many diverse approaches. | The conclusions will be revised to reflect the discussion from the relevant cases studies that will be presented. |

**Appendix, further detail on the proposed four case studies:**

**Case Study I**

In this case study, a non-governmental organisation would like to use soil information to design of a further round of feeding trial experiments to evaluate the potential of agronomic biofortification as a strategy `at scale' with local agronomic fortification. The non-governmental organisation will use the existing data from GeoNutrition to address that question, so no new sampling is to be undertaken. 
[revised manuscript text omitted]

| | |
|---|---|
| (U2) Can acceptable uncertainty be specified quantitatively? | On the above basis a critical probability might be elicited from a group, such that a community is considered for inclusion in the trial if P(Se$_{sol}$<Se$_{threshold}$) exceeds that critical value. Considerations in U1 suggest that the loss function is asymmetrical with a larger loss from false positives (when a community is initially engaged, and then removed from the trial) than false negatives (when feasible sites are not identified).  As noted by Chagumaira et al (2022) a threshold probability, $P_t$, for some condition (e.g. Se$_{sol}$<Se$_{threshold}$) at which to act as if it were true is equal to $\frac{L_1}{L_1+L_2}$ where $L_1$ is the loss incurred if that action is undertaken unnecessarily  (here the loss incurred by initiating a study at a site, then withdrawing after closer investigation), and $L_2$ is the loss incurred through failing to intervene where it was necessary (here the loss incurred by rejecting a suitable site). This threshold probability is the one which minimizes the expected loss.  This is a reasonable decision criterion to be applied over multiple candidate sites. Under the expected asymmetry $L_1 > L_2$ and so $P_t >$ 0.5.  The threshold could be narrowed down by an elicitation with a stakeholder group, either directly (as done by Chagumaira et al., 2022) or indirectly by using a formal elicitation in which the stakeholders are asked to complete a decision/state matrix as in Figure 2, populating the cells with factors that would contribute to the cost of either error type.  They are then asked to give a value for the ratio of the costs $L_1/L_2$.  In this case we assume that the ratio was 3, so $P_t$ = 3/4 =0.75. |
| (U3) Can survey effort be linked to the uncertainty measure? | Not relevant for a secondary data analysis as no new sampling is to be undertaken. |

*Preliminary results for Case Study I*

[Figure]

Figure 1 Probability that Se_sol concentration does not exceed 1.49 µg/kg expressed on a numerical scale for case study I. The area recommended for interventions is 5589 km$^2$

**Case Study II**

This end-user is very similar to that for case study I. However, their attitude to risk is different, for reasons which will be explained in the context of the framework, and as a result they are prepared to accept a larger probability that it will be necessary to withdraw from a proposed trial site if it is found to be unsuitable in exploratory investigations. An elicited threshold probability to delineate where possible trial sites occur is therefore smaller than in case study I, and so a larger region is delineated.

*Preliminary results for Case Study II*

[Figure]

Figure 2 Probability that Se$_{sol}$ concentration does not exceed 1.49 µg/kg expressed on a numerical scale for case study II. The area recommended for interventions is 9964 km$^2$

**Case Study III**

This end user is planning an intervention within a district of Malawi and wants a map of the district to support this. They want to investigate the sampling density required to produce information of a certain quality to plan further sampling. Because they have a clear sense of the implications of decisions made under uncertainty, they want the final product to express uncertainty in probabilistic terms. A suitable metric for selecting a sample size is the probability, conditional on a location having soil Se below the threshold, that it is not mapped as such. This can be computed from the existing data distribution and its variogram.

*Preliminary results for case study III*

The plot below shows the output. It is noted that the probability of a site where an intervention is needed being incorrectly mapped is reduced with increasingly dense sampling. The end-user decides in this context that a 1-km grid will suffice, bringing this probability close to 0.5, and with small marginal benefit for denser sampling, and

that a measure of the probability that the intervention is needed will be a necessary uncertainty measure for decision making from the final product.

[Figure]

Figure 3 Conditional probability that a site is *not* mapped as requiring an intervention, given that it does, plotted against grid spacing (note log scale) for Se$_{sol}$ concentration.

**Case Study IV**

This end user is planning further studies on soil and crop Se within a district of Malawi and wants a map of the district to support this. They want to investigate the sampling density required to produce information of a certain quality in order to plan further sampling. They do not have specific decisions to support but want a map that they can assume gives a picture of soil Se variation which is reliable in the sense that it can be regarded as repeatable and not dependent on arbitrary variations in the origin of the sample grid. The offset correlation statistic (a published criterion, which will be described in the revision) is suitable for this. An absolutely repeatable mapping procedure has an offset correlation of 1.0, and grid spacing determines the statistic, given the variogram.

*Preliminary results for case study II*

The plot below shows the output. Note that the offset correlation declines as sampling density increases. However, it exceeds 0.95 with a grid spacing of 5km, and this is deemed suitable by the end user.

[Figure]

Figure 4 Offset correlation plotted against plotted against grid spacing (note log scale) for Se$_{sol}$ concentration. The stakeholders will select the smallest offset correlation that would be acceptable if the map were to be used for their particular purpose.